# Biological condensates form percolated networks with molecular motion properties distinctly different from dilute solutions

Zeyu Shen[1], Bowen Jia[1], Yang Xu[1], Jonas Wessén[2], Tanmoy Pal[2], Hue Sun Chan[2], Shengwang Du[3,4†], Mingjie Zhang[5,6]*

[1]Division of Life Science, Hong Kong University of Science and Technology, ClearWater Bay, Kowloon, Hong Kong, China; [2]Department of Biochemistry, University of Toronto, Toronto, Canada; [3]Department of Physics, Hong Kong University of Science and Technology, Clear Water Bay, Kowloon, Hong Kong, China; [4]Department of Chemical and Biological Engineering, Hong Kong University of Science and Technology, Clear Water Bay, Kowloon, Hong Kong, China; [5]Greater Bay Biomedical Innocenter, Shenzhen Bay Laboratory, Shenzhen, China; [6]School of Life Sciences, Southern University of Science and Technology, Shenzhen, China

*For correspondence: zhangmj@sustech.edu.cn

Present address: [†]Department of Physics, The University of Texas at Dallas, Richardson, Texas, United States

Competing interest: The authors declare that no competing interests exist.

**Abstract** Formation of membraneless organelles or biological condensates via phase separation and related processes hugely expands the cellular organelle repertoire. Biological condensates are dense and viscoelastic soft matters instead of canonical dilute solutions. To date, numerous different biological condensates have been discovered, but mechanistic understanding of biological condensates remains scarce. In this study, we developed an adaptive single-molecule imaging method that allows simultaneous tracking of individual molecules and their motion trajectories in both condensed and dilute phases of various biological condensates. The method enables quantitative measurements of concentrations, phase boundary, motion behavior, and speed of molecules in both condensed and dilute phases, as well as the scale and speed of molecular exchanges between the two phases. Notably, molecules in the condensed phase do not undergo uniform Brownian motion, but instead constantly switch between a (class of) confined state(s) and a random diffusion-like motion state. Transient confinement is consistent with strong interactions associated with large molecular networks (i.e., percolation) in the condensed phase. In this way, molecules in biological condensates behave distinctly different from those in dilute solutions. The methods and findings described herein should be generally applicable for deciphering the molecular mechanisms underlying the assembly, dynamics, and consequently functional implications of biological condensates.

## Editor's evaluation

In this work, the authors introduce an adaptive single-molecule tracking approach for following molecules within biomolecular condensates. Consistent with the emerging idea that condensates are not simple, purely viscous, Newtonian fluids, the authors find that the motions of molecules reveal intrinsic inhomogeneities switching between trapped and more mobile states that suggest the existence of at least two – possibly more – states within condensates. The data appear to be consistent with the formation of percolated networks within condensates – a finding that is likely to be general to other systems.

## Introduction

Phase separation-mediated formation of condensed macro-molecular assemblies is being increasingly recognized as a general mechanism for cells to form a distinct class of cellular organelles with diverse functions (*Banani et al., 2017*; *Chen et al., 2020*; *Lyon et al., 2021*; *Shin et al., 2017*; *Wu et al., 2020*). Compared to the classical cellular organelles that are demarcated by lipid membranes, organelles formed via phase separation either do not associate with or are not enclosed by lipid membranes. Such organelles are referred to as biological condensates or membraneless organelles in the literature (we use biological condensates throughout this article). Formation of biological condensates greatly expands the means by which a living cell compartmentalizes its molecular constituents for specific and diverse functions. Since biological condensates are not enclosed by membranes, molecules within biological condensates can be in relatively fast dynamic exchange with their counterparts in dilute solution without energy input. This possibility constitutes an important basis for numerous unique properties of biological condensates, such as how sharp concentration gradient between the condensed and dilute phases is maintained, how molecules are selected to be included or excluded in the organelle compartment, the mechanisms for regulating organelle formation/ dispersion and the rates at which these processes can take place, etc., that are distinct from those of membrane-enclosed organelles (*Chen et al., 2020*; *Fare et al., 2021*; *Hyman et al., 2014*; *Lyon et al., 2021*; *Shin et al., 2017*). Research into the formation and function of biological condensate has gained extensive interest in recent years. Nonetheless, the burgeoning field is still in its infancy. Even fundamental physicochemical principles for the assembly of biological condensates, including the role of liquid–liquid phase separation, are under active investigations and vigorous debates (*McSwiggen et al., 2019*; *Mittag and Pappu, 2022*; *Musacchio, 2022*; *Pappu et al., 2023*).

Molecules within biological condensates can be massively concentrated. For example, proteins can be concentrated by more than 10,000 folds upon chromatin condensate formation (*Gibson et al., 2019*). In cell peripherals such as synapses in neurons, phase separation can concentrate numerous proteins into postsynaptic densities by >1000-fold (*Zeng et al., 2018*). A fundamental task in biological phase separation research is to understand how molecules in the condensed phase behave and function. The existing biochemistry and biophysics theories that have been guiding our understandings of molecular behaviors and their interactions in living cells for many years in the past are mainly developed for molecules in dilute solutions (*Cantor and Schimmel, 1980*). A biological condensate formed via phase separation is more of a condensed soft matter system, thus theories dealing with dilute solutions are not expected to be generally adequate for condensed molecular systems. Due to extreme complexities of molecular constituents (i.e., proteins, nucleic acids, and lipids), molecular compositions (i.e., each functional biological condensate often contains hundreds or more different types of molecules), and broad range of interaction modes (e.g., very large dynamic ranges of binding affinities and molecular valency, different levels of cooperativities, etc.) of biological condensates in cells, currently available theories in soft matter physics and polymer chemistry, though extremely useful, are likely not sufficient to be directly adapted to characterize biological condensates in a simplistic manner. Accordingly, extensive efforts are being made to better understand physics principles underlying the formation of biological condensates. Quantitative approaches have been developed to study properties of biological condensate in vitro and in living cells (see *Abyzov et al., 2022*; *Hubatsch et al., 2021*; *Song et al., 2020*; *Taylor et al., 2019* for a few examples). Super-resolution and single-molecule tracking-based methods are used to characterize the biophysical properties of biological condensates (*Cho et al., 2018*; *Garcia et al., 2021*; *Guilhas et al., 2020*; *Kent et al., 2020*; *Moon et al., 2019*; *Muñoz-Gil et al., 2022*). Theoretical and computational studies have also provided valuable insights into physical principles governing biopolymer-mediated condensate formation (*Berry et al., 2018*; *Bertrand and Lee, 2022*; *Bremer et al., 2022*; *Choi et al., 2019*; *Cinar et al., 2019*; *Espinosa et al., 2020*; *Farag et al., 2022*; *Feric et al., 2016*; *Lin et al., 2016*; *Zhou, 2021*), including—but not limited to—approaches addressing how the phase separations of disordered protein regions are governed by their amino acid sequences (*Das et al., 2020*; *Dignon et al., 2018*; *Harmon et al., 2017*; *Joseph et al., 2021*; *Lin et al., 2018*; *Lin et al., 2022*; *McCarty et al., 2019*; *Tesei et al., 2021*; *Wessén et al., 2022*).

Against this backdrop of exciting recent advances, here we address several foundational aspects of biological condensates' assembly and dynamics by developing an adaptive super-resolution imaging-based method that can simultaneously and robustly monitor and quantify motion properties

of individual molecules in dilute and condensed phases of biological condensates formed in solution or on lipid membranes. In addition to directly visualizing motion trajectories within and between phases, this method affords direct measurements of diffusion parameters of each molecule in the dilute and condensed phases. Unexpectedly, we observed that molecules in the condensed phase spend a very large fraction of time in transient motion-frozen state. Such temporary motion freeze exists in various biological condensates and is likely a consequence of the presence of specific and multivalent interaction-mediated large molecular network formation (i.e., percolation) in condensed phases. This observation is of fundamental significance because it is intuitively clear that motion property changes due to formation of biological condensates can fundamentally alter action mechanisms and cellular functions of biomolecules.

## Results

### Localization-based super-resolution imaging of phase separation

In an earlier study, we introduced a localization-based single-molecule tracking experiment to study motion properties of proteins in the condensed phase of in vitro reconstituted active zone condensates formed on two-dimensional supported lipid bilayer (SLB) (*Wu et al., 2019*). Here, we further developed the method into an assay that can simultaneously track molecules in both condensed and dilute phases. We used the in vitro reconstituted postsynaptic density (PSD) condensates formed on SLB (*Zeng et al., 2018*) to demonstrate this method. Four major PSD proteins (PSD95, Shank3, GKAP, Homer) and Trx-tagged GCN4-His$_8$-NR2B-CT tetramer (termed as NR2B in this article) were included in our study (*Figure 1A*). These five proteins, via specific and multivalent interactions, form a large molecular network capable of phase separation at physiological concentrations (*Zeng et al., 2018*).

Since the densities of proteins are hugely different between condensed and dilute phases, sparse labeling would lead to a lack of information for molecules in the dilute phase and dense labeling would cause extensive overlapping of single-molecule signals in the condensed phase during conventional fluorescence imaging experiments. To overcome this dilemma, we utilized dSTORM imaging (*van de Linde et al., 2011*) to obtain a large number of stochastically emitted single-molecule tracks in both condensed and dilute phases by sparse labeling proteins with photo-switchable dyes (in this case by labeling NR2B with 1% Alexa 647). TIRF illumination mode was used to detect protein signals on SLB so that signals from molecules not tethered to the membrane were minimized.

The PSD mixtures formed noncircular condensed phase on SLB with around or less than 1 µm in size, but conventional TIRF images could only provide fuzzy phase boundaries at this scale (*Figure 1B*, top; see also [*Zeng et al., 2018*]). The same area was then first photo-bleached by a high laser intensity and then imaged with a moderate laser intensity optimized for the fluorophore lifetime lasting for 3000 frames with an exposure time of 30 ms per frame, resulting in a high-resolution image containing ~100,000 individual localizations (*Figure 1B*, bottom). The overall phase boundary did not undergo obvious change during the imaging process as the shapes and boundaries of the condensed droplets in the system remained essentially the same (i.e., the stacked single-molecule images in the lower panel can be nicely superimposed to the TIRF image acquired at the beginning of the imaging session; see *Figure 1—figure supplement 1A*).

Due to the stochastic nature of fluorophore switch on and off, the reconstructed super-resolution image could be treated as static molecular distributions of labeled molecules in both dilute and condensed phases. Based on this super-resolution image, we could define those areas that have higher localization densities as condensed phase regions, and the rest as dilute phase regions (*Figure 1C*). The resulting phase boundary could be clearly visualized for each condensed region. Comparing the average localization densities in the condensed and dilute phases, we could estimate the partition coefficient of 60.8 ± 1.6 for NR2B (i.e., NR2B was enriched into the condense phase by ~61-folds). The calculated NR2B enrichment derived from the super-resolution imaging study was close to the value obtained by a bulk fluorescence imaging-based method shown in our previous study (*Zeng et al., 2018*). We noted with interest that the distribution of NR2B in the condensed phase is not homogeneous (*Figure 1C*), indicating possible formation of nanodomain-like clusters within the condensed phase. The existence of nanodomain-like NR2B clusters in the condensed phase is supported by auto-correlation function analysis of NR2B distributions (*Tang et al., 2016*; *Figure 1—figure supplement 1B and C*).

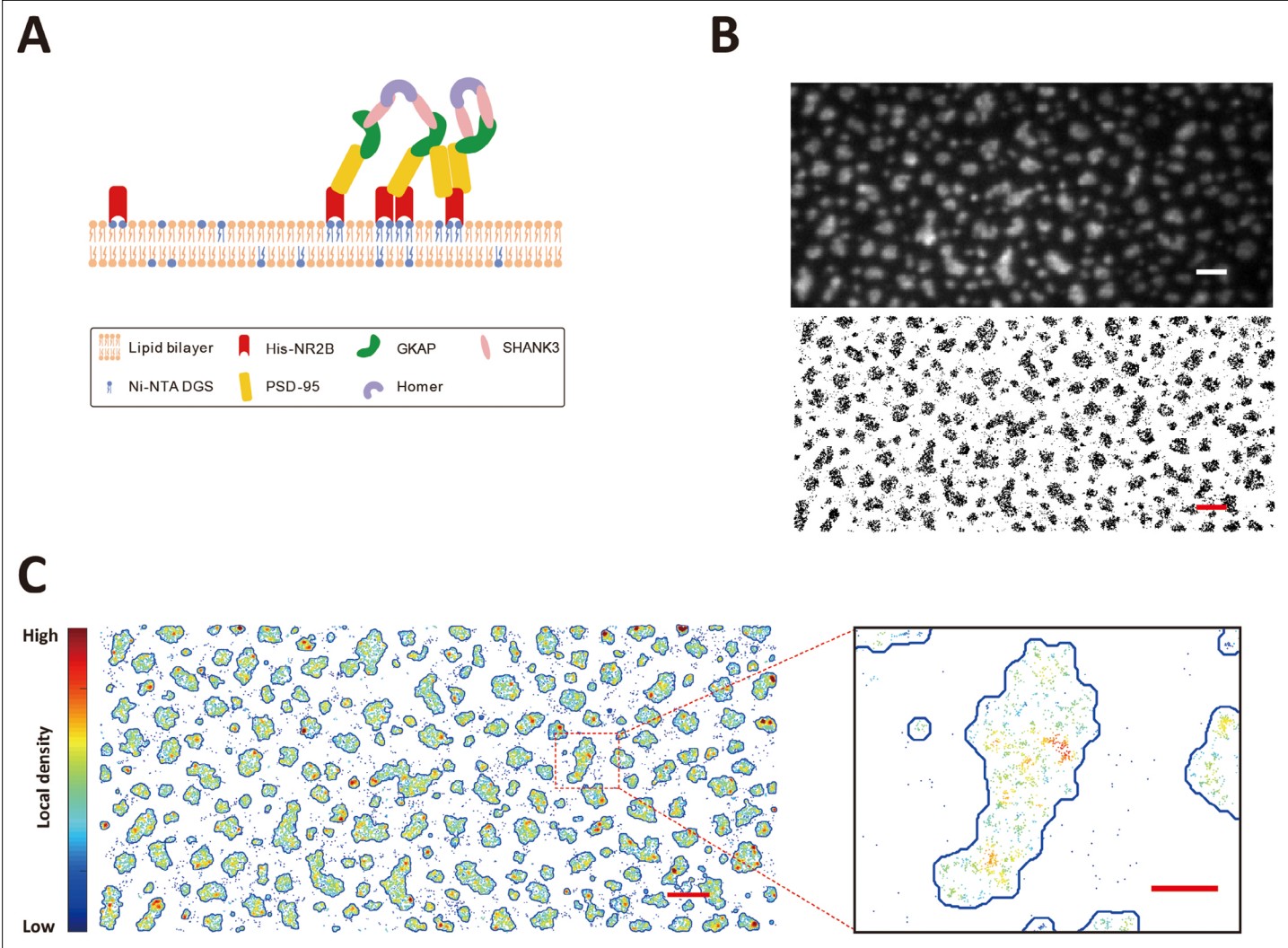

**Figure 1.** Single-molecule imaging of phase separation on supported lipid bilayers. (**A**) Schematic diagram showing phase separation of postsynaptic density (PSD) protein assembly on supported lipid bilayers (SLBs) (*Zeng et al., 2018*). (**B**) Upper panel: a TIRF image of Alexa 647 labeled His$_8$-NR2B tetramer clustered within the PSD condensate on SLB. Lower panel: stacking of 4000 frames of dSTORM images of Alexa 647 labeled His$_8$-NR2B within the same PSD condensates as shown in the TIRF image above. Black dots represent localizations recognized during the imaging. Scale bar: 2 μm. (**C**) Phase boundary of the PSD condensates determined by localization densities. The boundaries are shown by blue lines. Localizations are color-coded according to their local densities from low (blue) to high (red). A zoom-in view of a typical condensed patch on SLB showing heterogeneous distributions and nano-cluster-like structures of molecules within the condensed phase. Scale bar of the original image: 2 μm, scale bar for the zoom-in view: 500 nm.

The online version of this article includes the following figure supplement(s) for figure 1:

**Figure supplement 1.** Determination of boundaries of the condensed phase throughout the imaging process and heterogeneous distribution of molecules in the condensed phase.

## Simultaneous single-molecule tracking in different phases

The localizations obtained from dSTORM images contained information about distributions as well as mobilities of molecules in both phases. However, the diffusion mode and densities of molecules are very different in condensed and dilute phases. A striking feature is that molecules tend to experience transient confinement in the condensed phase (*Video 1*). We developed an adaptive single-molecule tracking algorithm that could automatically and robustly define optimal search ranges for molecules in different phases, and the method could effectively minimize the global assignment errors in tracking molecules in both condensed and dilute phases (*Figure 2A* and *Figure 2—figure supplements 1–4*; see 'Materials and methods' for extended description of the algorithm). Briefly, phase boundaries were determined initially by densities of localizations. A default search range (500 nm) was used to assign

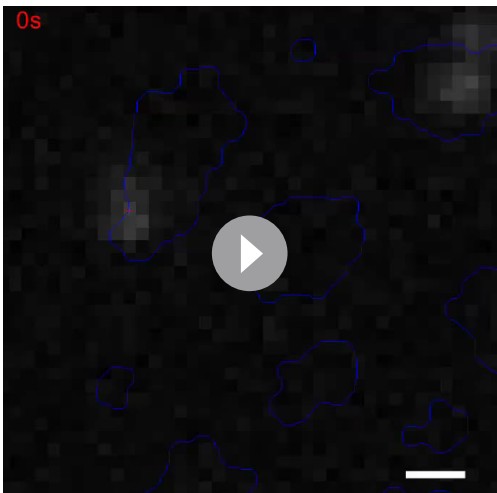

**Video 1.** Raw image superimposed with phase boundary and tracks (steps length >5) in the NR2B + postsynaptic density (PSD) phase separation system on 2D supported lipid bilayer (SLB). Red lines represent tracks in the condensed phase, green lines represent tracks in the dilute phase, and yellow lines represent tracks crossing phase boundaries. Molecules can be seen to switch between a confined state and a mobile state. Molecules are also directly observed to diffuse across phase boundaries. This video is played in real time. Scale bar: 500 nm.

https://elifesciences.org/articles/81907/figures#video1

all localizations into tracks in both condensed and dilute phases. Diffusion coefficients of NR2B in both dilute and condensed phases were estimated for determining optimized search range for different phases in subsequent analyses. The initial roughly estimated overall diffusion coefficients of NR2B in condensed phase and dilute phase are 0.22 $\mu m^2$/s and 0.57 $\mu m^2$/s, respectively (see analyses below). All localizations were reassigned with the optimized search range to obtain final tracks of NR2B in both condensed and dilute phases.

After adaptively assigning all localizations into tracks, we could obtain single-molecule tracks of molecules in both condensed and dilute phases (*Figure 2B*). With this method, we could directly record events of molecules entering into and escaping from the condensed phase as well as switch motions of molecules converting between a confined state and a mobile state in the condensed phase (*Figure 2C*). The number of NR2B molecules entering into and escaping from the condensed phase was equal (*Figure 2D*), a finding that is consistent with the bulk equilibrium state of the PSD condensate. Interestingly, NR2B molecules within the condensed phase spent a large proportion of time in the confined state and molecules could switch between the confined state and the mobile state (*Figure 2C and E*, *Video 1*). No confined state could be detected when only NR2B was tethered to SLB (*Figure 3—figure supplement 1A*). The above result indicated that NR2B in the condensed phase did not undergo homogeneous diffusion motions as one might expect for molecules in certain crowded condensed phases (*Condamin et al., 2008*; *Höfling and Franosch, 2013*; *Woringer and Darzacq, 2018*). Instead, the motions of NR2B in the condensed phase resemble previously studied crowded and confined systems with transient caging/trapping dynamics (*Akimoto et al., 2011*; *Bhattacharjee and Datta, 2019*; *Weeks and Weitz, 2002*; *Wong et al., 2004*). Comparisons with data from a set of model experimental systems of FUS-Shank3 chimeric proteins (see Figure 4) suggest that the transient kinetic trapping of NR2B (transiently remaining very close to a stationary position, i.e., very slow diffusion) in the PSD condensed phase is likely due to its relatively strong binding to certain parts of a percolated PSD network with slow dynamics. Accordingly, NR2B molecules in the condensed phase are expected to undergo subdiffusive motions. Indeed, our single-molecule tracking data of NR2B motion in the condensed PSD phase (see *Figure 2E* for an example) fits well with typical behaviors of subdiffusion (see below and *Berezhkovskii et al., 2014*; *Bouchaud and Georges, 1990*; *Condamin et al., 2008*; *Netz and Dorfmüller, 1995*; *Saxton, 2007*). We also imaged motions of NR2B in the PSD condensates formed in 3D solution at single-molecular resolution and found that, in the condensed phase, each NR2B molecule spent a large proportion of time in the confined state in a manner similar to that for the 2D case (*Figure 3—figure supplement 1B*).

The histogram of NR2B displacement tracks in the condensed PSD phase has a dominant peak with very small displacements, corresponding to the large proportion of time NR2B spends in a confined state or an ensemble of confined states. In contrast, a small and relatively flat shoulder tailing the main peak represents the small proportion of time NR2B is in a mobile state or an ensemble of mobile states moving with larger displacements during the same given time interval (*Figure 3A1*). Fitting the histogram with a single population of NR2B undergoing Brownian motion could only cover the confined state peak of molecule but not the high-displacement tail of this distribution (*Figure 3—figure*

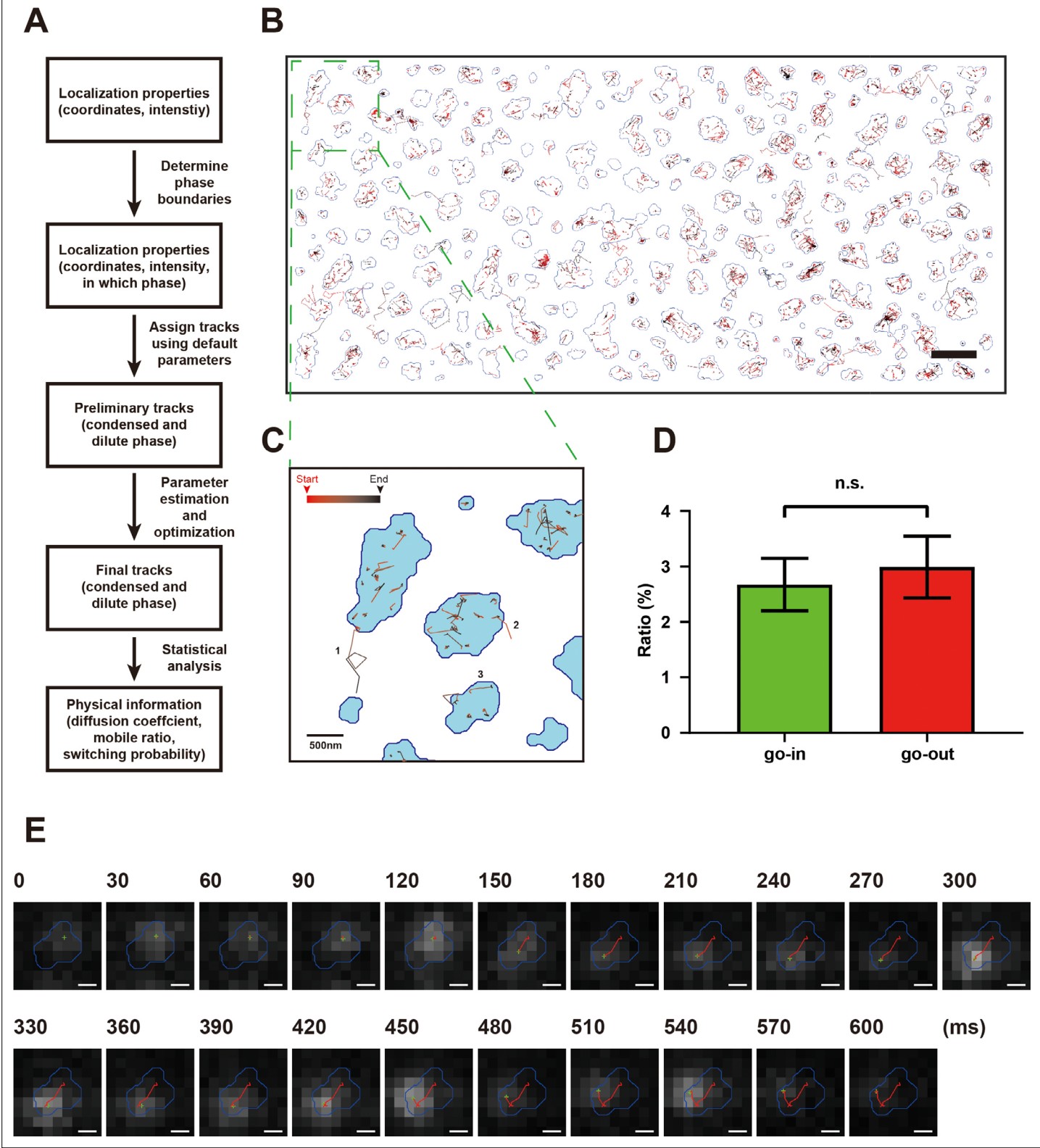

**Figure 2.** Development of an adaptive single-molecule tracking algorithm for imaging single molecules in the condensed and dilute phases simultaneously. (**A**) Flowchart of the adaptive single-molecule tracking algorithm. (**B**) Assignments of motion tracks of NR2B in both condensed and dilute phases in the postsynaptic density (PSD) condensates formed on supported lipid bilayer (SLB). Each track is color-coded from red to black from the beginning to the end of the track. The boundaries of the condensates are marked by blue lines. Scale bar: 2 μm. (**C**) Representative tracks showing

*Figure 2 continued on next page*

*Figure 2 continued*

typical NR2B motions. Examples include trajectories involving mobile and/or confined states as well as exchange events of molecules crossing between condensed and dilute phases (tracks 1 and 2) and a trajectory that passes through both the transiently confined and the mobile (more freely diffusing) states in the condensed phase (track 3). (**D**) Percentages of NR2B molecules exchange from the dilute phase into the condensed phase ('go-in') and vice versa ('go-out') were counted in four sessions of dSTORM imaging experiments. No significant difference between go-in ratio and go-out ratio was detected. p=0.378 using paired *t*-test and defined as not significant (n.s.). (**E**) Raw image data superimposed with phase boundary (blue line), molecule localization (green cross), and track steps (red line) show a typical trajectory of an NR2B molecule undergoing multiple motion switches between confined and mobile states in the condensed phase. Scale bar: 200 nm.

The online version of this article includes the following figure supplement(s) for figure 2:

**Figure supplement 1.** Evaluation of the adaptive single-molecule tracking algorithm by simulation and by experiments.

**Figure supplement 2.** Simulations of track assignment errors of phase separations with molecules in the condensed phase undergoing homogeneous free diffusions.

**Figure supplement 3.** Simulations of diffusion coefficient errors of phase separations with molecules in the condensed phase undergoing homogeneous free diffusions.

**Figure supplement 4.** Simulations of track assignment errors of phase separations with molecules in the condensed phase containing both confined and mobile states.

*supplement 1C1*). By comparison, in the dilute phase, the overall displacements of NR2B are much larger as well as broader, and there is no prominent peak at very small displacements (*Figure 3A2*). Nonetheless, the displacement histogram of NR2B in the dilute PSD phase still cannot be described entirely by simple Brownian diffusion of a homogeneous NR2B population (*Figure 3—figure supplement 1C2*), though a simple Brownian diffusion is a better approximation in this case, suggesting a likely presence of multiple populations/states of pre-percolated NR2B/PSD protein complexes even in the dilute phase, a feature that would be in line with recent observations that some subsaturated solutions of condensate-forming biomolecules are not featureless (*Kar et al., 2022*). However, in a control system with only NR2B tethered to SLB (i.e., no addition of any other PSD proteins), the histogram of NR2B displacements can be reasonably fitted by a simple diffusion model (*Figure 3B*).

To gain further physical insight into the dynamic behaviors of NR2B in our PSD/SLB system, we first applied a Hidden Markov Model (HMM) to fit the motions of NR2B in the condensed phase with a simple two-state diffusion model (*Das et al., 2009*; *Persson et al., 2013*). This initial analysis is motivated fundamentally by the clear observation of a dichotomy between the coexisting relative immobile NR2B and mobile NR2B molecules in the imaging experiments (*Figure 2E*, *Video 1*). In this approach, the model parameters to be extracted include the diffusion coefficients for presumed simple diffusion of NR2B molecules in transiently confined and mobile states ($D_c$ and $D_m$) and the switching probabilities between the two states ($P_{mc}$ and $P_{cm}$). Maximum likelihood methodology was used to estimate the parameters iteratively, and the parameters converged quickly after several thousand iterations of optimization (*Figure 3C and D*). The resulting best-fit diffusion coefficients in the presumed mobile and confined states were, respectively, $D_m = 0.167 \pm 0.002$ µm²/s and $D_c = 0.0127 \pm 0.0001$ µm²/s. The switching probabilities between these presumed states were $P_{mc} = 82.8\% \pm 0.3\%$ (mobile to confined states) per frame and $P_{cm} = 3.8\% \pm 0.1\%$ (confined to mobile states) per frame. The confinement ratio ($P_c$) in this type of models, defined as the percentage of time that a molecule spends in the confined state, could in general be calculated in the model by the two switching probabilities as $P_c = P_{mc}/(P_{mc} + P_{cm})$. For NR2B in the present model for PSD condensates, $P_c = 95.6\%$. The mobile ratio, defined as the percentage of time that a molecule spends in the mobile state, could then be calculated as $P_m = 1-P_c = 4.4\%$. Assuming further that molecular motion in the dilute phase may be described as a first approximation by a simple diffusion process, a diffusion coefficient of the molecule in the dilute phase could be estimated by fitting the mean square displacement (MSD) values as a function of time (i.e., using the equation MSD = $4Dt$ based on simple diffusion for the fitting). At the same time, an overall generalized diffusion coefficient for NR2B in the condensed phase may also be estimated by fitting the MSD with a subdiffusive model of MSD = $4D_\alpha t^\alpha$ (*Metzler et al., 2014*). The fitted diffusion coefficient of presumed simple diffusion of NR2B in the dilute phase of the present PSD condensate is $0.47 \pm 0.12$ µm²/s, which is close to that of NR2B alone tethered to SLB ($0.61 \pm 0.04$ µm²/s) (*Figure 3E*). The quantitative similarity of these two estimated diffusion coefficients is particularly striking in view of the much lower diffusion rate reflected by the overall generalized diffusion coefficient for subdiffusion of NR2B in the PSD condensed phase ($D_\alpha = 0.014 \pm 0.001$ µm²/s$^{0.74}$ with $\alpha$

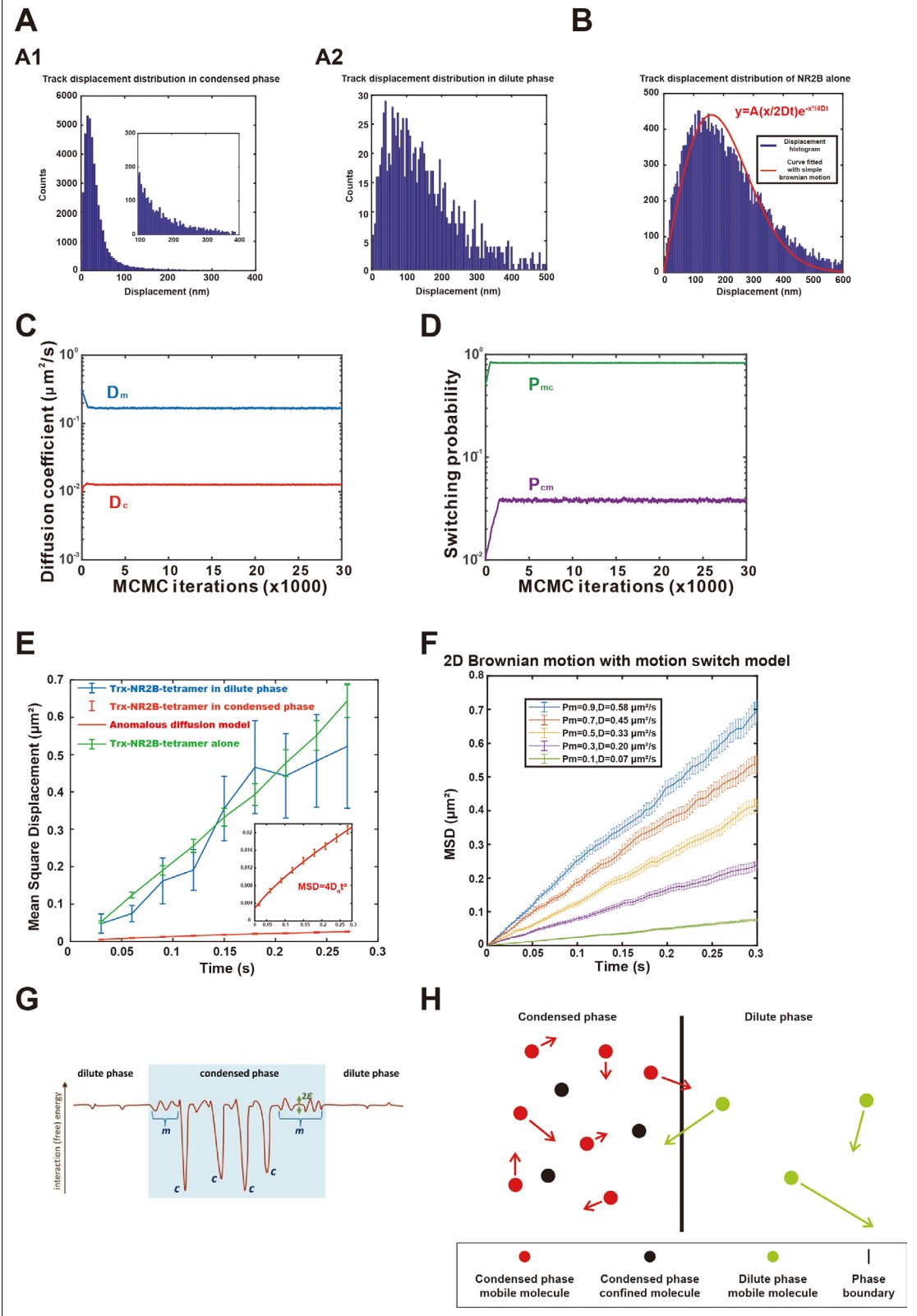

**Figure 3.** Dynamic parameters and a diffusion model for an equilibrium state phase separation system. (**A**) Displacement distribution of tracks in the (**A1**) condensed and (**A2**) dilute phases. Zoom-in view in (**A1**) shows detailed distribution of the distribution tail. Bin size of histogram is 5 nm. (**B**) Displacement distribution of tracks of NR2B alone tethered to supported lipid bilayer (SLB). Red curve is the fit with a simple 2D Brownian motion distribution obtained by nonlinear least-squares method using MATLAB, here the fitted diffusion coefficient $D = 0.46 \pm 0.02$ μm²/s, square of Pearson

*Figure 3 continued on next page*

*Figure 3 continued*

correlation coefficient $r^2 = 0.91$, root mean square deviation (RMSE) = 42.5. Bin size of histogram is 5 nm. (**C, D**) Optimization of the dynamic parameters of NR2B in the postsynaptic density (PSD) condensates formed on SLBs with Hidden Markov Model assuming that NR2B is conforming to a two-state motion model in the condensed phase visiting both a transient confined state and a mobile state. The parameters of interest are diffusion coefficient of the confined state in the condensed phase ($D_c$) and diffusion coefficient of the mobile state ($D_m$) in the condensed phase, switching probability from the confined state to the mobile state ($P_{cm}$) and the reversed switching probability ($P_{mc}$). (**E**) Determination of the diffusion coefficients of NR2B in dilute phase (blue) by fitting the mean square displacements (MSDs) against time using linear regression. In view of the MSD's appreciable nonlinear time dependence, motion of NR2B in the condensed phase (red) is fitted by a subdiffusive model, viz., MSD = $4D_\alpha t^\alpha$. The figure also includes data and linear fit for the control case of NR2B alone tethered to SLB (green). The inset shows a y-axis zoom-in view of NR2B dynamic behavior in the condensed phase to highlight its subdiffusive nature. The number of trajectories used in the fittings was 2443 for the condensed phase, 13 for the dilute phase, and 248 for NR2B alone on SLB. Note that the NR2B-alone diffusion coefficient 0.61 ± 0.04 μm²/s obtained by MSD fitting is similar in value but not identical to that obtained in (**B**) by fitting displacements distribution, underscoring that the underlying process is not exactly a simple Brownian diffusion. (**F**) Monte Carlo (MC) simulation of molecular diffusion on a 2D surface in the motion switch model under different mobile ratio ($P_m$), apparent diffusion coefficient (*D*) was obtained by fitting the MSD curve. As an example, the diffusion coefficient in mobile state is chosen to be $D_m = 0.61$ μm²/s (same as what we measured experimentally for NR2B alone tethered to SLB, green line in **E**), whereas $D_c = 0$ and $P_{cm} = 0.1$ for all $P_m$ considered; see 'Materials and methods' for details. Note that subdiffusion is not well captured by this simple MC model; fitted *D* values here are those for the simple diffusion model ($\alpha = 1$). (**G**) Schematic of a physically plausible energy landscape experienced by a given NR2B molecule. Here the vertical axis represents interaction free energy (in units of Boltzmann constant times absolute temperature) with the other PSD biomolecules (including interactions with other NR2B molecules but not water molecules, including entropic effects of aqueous solvation but not translational entropy of the given NR2B molecule) and the horizontal axis represents schematically the multiple spatial coordinates of the given NR2B molecule. Because of the liquid-like/soft-matter nature of the PSD system, such a landscape should change with time but the following salient features are expected to be robust. In the condensed phase (blue-shaded region), confined states are envisioned to be underpinned by the deep wells (labeled '*c*') for strongly favorable interactions, whereas mobile-state diffusion are seen to be affected by small variations between repulsive and attractive interactions (labeled '*m*') with typical ranges of $2\varepsilon$. Weak interactions can also occur in the dilute phase but more rarely (small bumps and dips in the unshaded dilute region). (**H**) Schematic diagram showing molecular motions between condensed phase and dilute phase under steady-state equilibrium conditions in accordance with the approximate empirical relation given by *Equation 1* for the present PSD system. Black and red dots represent, respectively, molecules in the confined and mobile states of the condensed phase. Green dots represent molecules in the dilute phase. The lengths of the arrows indicate different mobilities of molecules in different phases. The extremely low mobility of molecules in the confined state are not indicated ($D_c/D_m \ll 1$).

The online version of this article includes the following figure supplement(s) for figure 3:

**Figure supplement 1.** Typical tracks of NR2B only tethered to supported lipid bilayer (SLB) or NR2B in 3D postsynaptic density (PSD) condensates.

**Figure supplement 2.** No obvious hindrances against motions when molecules cross the phase boundaries.

**Figure supplement 3.** Correlation-based classification of molecular displacements without presuming simple diffusion.

= 0.74 ± 0.03; see *Figure 3E*). The generalized diffusion coefficient $D_\alpha$ contains different contributions from both confined and mobile states of the NR2B molecules in the condensed phase (as manifested by the fact that $\alpha \neq 1$). When the confinement ratio is large, this generalized diffusion coefficient will naturally be dominated by slowing-down effects of the confined state and therefore significantly differ from the diffusion coefficient in the mobile state. The expected general trend of variation of overall diffusion speed with confinement ratio on 2D SLB is illustrated by the Monte Carlo simulation results for the simple two-state model in *Figure 3F*.

## Physical picture and a two-state, two-phase diffusion model for equilibrium and dynamic properties of PSD condensates

The above model with HMM-estimated parameters is thus based upon three simple diffusion processes, namely, the confined- and mobile-state diffusion processes in the condensed phase and a single simple diffusion process in the dilute phase. To assess whether such a simple model, at least as a first approximation, can provide a reasonably adequate description of our experimental observations, we next developed a corresponding physical model based on multiple simple diffusion processes to describe a phase separation system based on the aforementioned parameters estimated by HMM from our adaptive single-molecule tracking data. Intuitively, in view of the experimental evidence for two vastly different time scales for condensed-phase dynamics as characterized by the motions of NR2B molecules, the energy landscape experienced by an NR2B molecules is expected to exhibit features depicted schematically in *Figure 3G*. In this tentative physical picture, confined states are associated with deep energy wells, whereas mobile-state molecular motions are affected by smaller variations in both attractive and repulsive interactions (range of variations ~$2\varepsilon$, with $\varepsilon \ll$ confined-state well depths). Owing to the inherent complexity of condensed phase structure and energetics,

these wells are envisioned to have different depths in general, thus the properties of the 'confined' and 'mobile' states are understood to be ensemble averages. Under appropriate conditions wherein the condensed-phase mobile-state potential energy variations are approximately equally divided between attractive and repulsive interactions relative to the dilute-phase baseline (as depicted in *Figure 3G*), a simple relationship between steady-state equilibrium condensed mobile-state and dilute-phase populations (governed by Boltzmann factors $\exp(\pm\varepsilon)$) and the effective diffusion coefficients in the two populations can be obtained from a correspondence between the Kawasaki acceptance criteria for Monte Carlo moves ($\sim\exp(\pm\varepsilon)$) and dynamic behaviors governed by the Smoluchowski equation for diffusion on a potential surface with position-dependent interaction energy (*Zhang and Chan, 2012*). When applied to the NR2B molecules in the present PSD system with condensed mobile-state diffusion coefficient $D_m$, dilute-phase diffusion coefficient $D_d$, and molecular (NR2B) densities $\sigma_c$ and $\sigma_d$ in the condensed and dilute phases respectively, such a consideration indicates that the following relationship holds approximately between the kinetic diffusion coefficients and the equilibrium enrichment fold (EF) of molecules in the condensed phase over the dilute phase:

$$\mathrm{EF} \approx \sigma_c/\sigma_d = D_d/\mathrm{P_m}D_m \qquad (1)$$

or

$$\text{Enrichment fold} \approx \frac{\text{Diffusion coefficient in dilute phase}}{\text{Diffusion coefficient in condensed phase of mobile molecules} \times \text{Mobile ratio}}$$

where EF is defined as $\sigma_c/\sigma_d$, and $P_m$ is the mobile ratio of molecules in the condensed phase. For NR2B in the PSD condensates formed on SLB, the diffusion parameters estimated by HMM from experimental measurements are $D_d = 0.47~\mu m^2/s$, $D_m = 0.17~\mu m^2/s$, and $P_m = 4.4\%$ as stated above. It follows that the EF of NR2B in the condensed phase estimated via the approximate relation in *Equation 1* by using these experimental kinetic parameters is $D_d/P_m D_m = 62.8$. Since this kinetics-estimated EF is very close to the equilibrium value of ~61 derived independently from experimentally observed localizations (*Figure 1B*), we regard *Equation 1* as empirically validated for the present PSD system, although the relationship between kinetic and thermodynamic properties of diffusive systems (*Berry et al., 2018*) can be rather complex in general for rugged energy landscapes (*Banerjee et al., 2014*; *Zwanzig, 1988*). It follows that the above physical picture based on three simple diffusion processes with a confined and a mobile state for the condensed phase does provide a viable rationalization for our experimental tracking data. In our analysis below, the simple relation in *Equation 1* serves to connect the microscopic kinetic variables (diffusion coefficient and mobile ratio) to macroscopic observables (enrichment fold), allowing for the simulation of large differences in molecular density between condensed and dilute phases without explicit consideration of free energy and Boltzmann factors. Consider a small region near the phase boundary (*Figure 3H*). The dilute phase contains sparse and fast-moving molecules. The condensed phase contains dense and slow-moving molecules that can be further categorized into either in a mobile state or a transiently confined state. Because we did not observe any obvious hindrance against motions when molecules cross the phase boundaries (*Figure 3—figure supplement 2*), the energy barrier at the interface between the condensed and dilute phases is likely not large in this system (*Brangwynne et al., 2011*; *Feric et al., 2016*), though diffusion barriers at phase boundaries were reported in other systems (*Peng and Weber, 2019*; *Strom et al., 2017*). The simple approximate relation in *Equation 1* then allows us to take the number of molecules crossing the boundary from one side to the other as being proportional to the diffusion coefficient ($D_m$ or $D_d$) and the molecular density ($\sigma_c$ or $\sigma_d$) of the initiating side such that the equilibrium density difference across the boundary can be maintained by the steady-state condition that the flux of molecules from the dilute phase to the condensed phase is equal to the flux in the opposite direction.

To assess the self-consistency of the above-described diffusion model, we conducted simulations of molecular motions from homogeneously mixed state toward a phase-separated equilibrium state by using a 2D simulation box of size $15 \times 30~\mu m^2$ with stationary phase boundaries from our experimental observation (*Figure 1B*) and periodical boundary conditions (*Figure 3—figure supplement 1D*). As described in 'Materials and methods' under the heading 'Equilibrium simulation of coexisting phases and confined/mobile states,' diffusion coefficients for the simulated molecules in the condensed phase ($D_m$ for mobile state only, confined-state molecules are assumed to be stationary for simplicity, i.e., $D_c \to 0$) and in the dilute phase ($D_d$), the mobile ratio, and the mobile state lifetime (derived from

**Table 1.** Simulation of phase separations with different input of diffusion parameters.

Monte Carlo method-based simulations of molecular diffusion on supported lipid bilayer (SLB) with experimental phase boundaries. A total of 50,000 molecules were included in each simulation, and these molecules were initially randomly distributed at the beginning of each simulation. Each simulation lasted for 100 s and was independently repeated 10 times. Simulated enrichment fold values were presented as mean value of last 10 s ± SD, where SD is standard deviation for the 10 independent runs.

| Input parameters | | | | Theoretical results | Output results |
|---|---|---|---|---|---|
| $D_m$ (µm2/s) | $D_d$ (µm2/s) | Mobile ratio | Mobile state lifetime (s) | Enrichment fold (EF) | Enrichment fold (EF) |
| 0.2 | 0.6 | 0.05 | 0.1 | 60 | 58.9 ± 0.7 |
| 0.1 | 0.6 | 0.1 | 0.1 | 60 | 58.6 ± 0.9 |
| 0.01 | 0.6 | 1 | - | 60 | 57.0 ± 0.8 |
| 0.1 | 0.6 | 0.1 | 0.5 | 60 | 57.9 ± 0.7 |
| 0.1 | 0.6 | 0.05 | 0.1 | 120 | 116.7 ± 1.9 |
| 0.2 | 0.6 | 0.1 | 0.1 | 30 | 29.5 ± 0.3 |

switching probability) of the molecule in the mobile and confined states in the condensed phase were used as simulation input (*Table 1*). A Monte Carlo algorithm was used to simulate the dynamics of a total of 50,000 molecules diffusing in the box for 100 s for each given condition. The EF between condensed and dilute phases along the simulation trajectory was determined accordingly. Every simulated system eventually reached equilibrium. As a test of the pertinent principles, we first considered a set of input parameters with expected EF values (from *Equation 1*) equal to 30, 60, or 120. The resulting steady-state EFs in the final equilibrium state under each simulated condition came very close to the kinetic-estimated value provided by the approximate relation in *Equation 1* (≤5% difference; see *Table 1*). While the precise reason for the small mismatches between expected and simulated remains to be ascertained, they likely arise from the irregular shapes of the experimental phase boundaries and therefore single Monte Carlo steps may traverse both the dilute and condensed phases even if the starting and end points of a step is in the same phase. When the experiment-based, HMM-estimated $D_d$ = 0.47 µm²/s, $D_m$ = 0.17 µm²/s, $P_{mc}$ = 0.828, $P_{cm}$ = 0.038 (per 30 ms), and thus $P_m$ = 4.4% values (see above) were applied to the same Monte Carlo setup, it yielded a simulated EF = 62.2 ± 0.3 (uncertainty based on 10 independent runs with different random seeds), which is practically identical to the EF value of 62.8 estimated by *Equation 1*. All in all, these test runs illustrate that our diffusion model is capable of providing a self-consistent physical picture.

In a broader context, if the molecules have multiple diffusion states and similar features of the underlying energy landscape of the coexisting phases are applicable, one may further extend the model, viz.,

$$\sum_i^m \sigma_i^\alpha D_i^\alpha \approx \sum_j^n \sigma_j^\beta D_j^\beta \qquad (2)$$

where molecules in phase α and β contain, respectively, m and n types of diffusion states. Under appropriate conditions, the approach can be further generalized to situations in 3D and/or with multiple (more than two) coexisting phases, in which case the present 2D molecule densities are replaced with corresponding 3D molecular concentrations and the sum of the products of molecule density and diffusion coefficient in different states should be equal for all the coexisting phases.

Our energy landscape picture provides further mechanistic insights into some unique properties of molecules in the condensed phase. For example, because mobile-state diffusion in the condensed phase is envisioned to be governed by a more rugged energy landscape than that in the dilute phase (*Figure 3G*), the effective diffusion coefficient for the mobile state in the condensed phase (which in principle can be derived from the underlying energy landscape) is expected to be smaller than that in the dilute phase. This is indeed the case experimentally. Nonetheless, although concentrations of molecules in the condensed phase are much higher than those in the dilute phase, the diffusion coefficient for the mobile fraction of molecules in the condensed phase may

not be dramatically different from that in the dilute phase ($D_m$ = 0.17 μm²/s vs. $D_d$ = 0.47 μm²/s), indicating that the landscape governing mobile-state diffusion in the condensed phase is only moderately rugged. In the two-state picture, it is obvious that the fraction of time that a molecule spends in the mobile state vs. the confined state can dramatically influence the macroscopic properties of the molecule in the condensed phase. For instance, in the present perspective based on two simple diffusion processes in the condensed phase, NR2B spends over 95% of the time in the confined state in the PSD condensate. Accordingly, the overall generalized diffusion constant of NR2B in the condensed phase is small at $D_\alpha$ = 0.014 μm²/s⁰·⁷⁴ instead of 0.17 μm²/s estimated for the mobile NR2B fractions in the condensed phase. Additionally, the EF of a molecule into the condensed phase upon phase separation is also dominantly reflected by the fraction of time molecules spent in the mobile state vs. that in the confined state. As we will demonstrate below experimentally, the binding affinity between molecules in the phase separation system and the molecular network complexity in the condensed phase determines the fraction of the time that a molecule spends in the mobile state vs. confined state as well as the enrichment of molecules in the condensed phase.

## Classification of molecular displacements without presuming simple diffusion models

It should be emphasized that while quantitative aspects of this two-state perspective of condensed-phase NR2B dynamics were initially deduced by using an HMM assuming two simple diffusion processes, the observation of a general separation of time scales into roughly a low-mobility (L) confined state and high-mobility (H) mobile state in the model PSD condensate is intuitive (*Video 1*) and can be quantified without presuming simple diffusion. This is showcased by a 'model-independent' formulation we developed here to classify experimental displacement steps based on a criterion on the correlation of the magnitudes of consecutive displacement steps in each experimental track. As detailed in 'Materials and methods,' we found that the magnitudes of displacements $d_i$ of consecutive steps (experimental time frames) in condensed-phase NR2B are correlated, indicating clearly the process is not simple Brownian diffusion. Analysis of conditional displacement distributions (subsets of overall distribution) based on putative demarcation values ($d_b$) for $d_i$ suggests that $d_b$ = 60 nm provides a reasonable classification of displacement steps into two broad classes, namely a low-mobility (L) state and a high-mobility (H) state that correspond roughly to the above HMM-deduced confined and mobile states (*Figure 3—figure supplement 3*). The dynamic complexity of the PSD condensed phase is underscored by the observation in *Figure 3—figure supplement 3D1 and D2* that the displacement distributions of both the L and H states exhibit deviations from simple Brownian diffusion, with the deviation more appreciable for the H states, which are better fitted by a Lévy distribution (*Viswanathan et al., 2008*; *Wang et al., 2020*). Nonetheless, as a first approximation, L-state and H-state motion can be reasonably fitted by two separate simple diffusion process to provide a semi-quantitative description of our experimental data. In this simplified framework, the confined (L) state distribution was interpreted as caused by detection error, whereas the mobile (H) state was described approximately by a simple Brownian diffusion process with diffusion coefficient $D_m$ = 0.044 ± 0.05 μm²/s (*Figure 3—figure supplement 3D*). As specified in 'Materials and methods,' the corresponding confined and mobile ratios and switching probabilities between the two motion states were readily obtained, with the mobile ratio $P_m$ = 13.5%. Together with the dilute-phase $D_d$ = 0.47 μm²/s estimated above, the EF estimated using *Equation 1* is EF ≈ $D_d$/ $P_m D_m$ ≈ 79. Applying these parameters (instead of HMM parameters) to the above-described Monte Carlo simulation using experimental phase boundaries yielded an EF value of 76.2 ± 0.6. In view of the aforementioned approximate nature of the fitted simple diffusion process for mobile-state motion and that the $D_m$ and $P_m$ values estimated from *Figure 3—figure supplement 3* are appreciably different from those estimated by HMM, the EF value of ≈76 from the present model-free, correlation-based consideration is seen as reasonably close to the experimental value of EF ≈ 61, thus attesting to the robustness of the physical picture that NR2B diffusion in the PSD condensates is roughly separable into two dynamic time scales. However, as stated above, the confined state and the mobile state each likely represents a combination of many substates due to complex molecular network formation in the condensed phase. The nature of these substates deserves further experimental investigations and more refined theoretical modeling in future studies.

In any event, taking all the above experimental data and computational analysis together, we now have a workable physical picture for molecular diffusion in the equilibrium state of a phase separation system. Our conceptual framework explicitly connects a set of measurable microscopic molecular motion properties with the observable macroscopic parameters of molecules in the system. The experimental method and the theoretical models developed above are robust and simple to implement. They should be useful for analyzing biomolecular phase separations in general.

## Dynamic molecular networks in condensed phases

Motions of molecules or molecular complexes in many dilute solutions are well described by simple diffusions. In sharp contrast, the present observations indicate that NR2B molecules in the PSD condensates formed on SLB and in 3D solution spend a very large proportion of time in the immobile/confined state, suggesting that PSD condensates form some sort of very large and thus essentially immobile molecular network (a process known as percolation in associative polymers including biopolymers; *Choi et al., 2020*; *Harmon et al., 2017*; *Winnik and Yekta, 1997*) capable of trapping NR2B transiently. Since molecular processes such as molecular interactions, chemical reactions, etc., require molecules to be able to collide with each other, phase separation-mediated immobilization of biomolecules in condensed phases can have huge implications on numerous fundamental properties of these molecules (e.g., binding kinetics, catalytic speed and specificity of enzymes, spatial distributions in cellular subcompartments, etc.).

Therefore, we next asked how such network-like structure might form and what factors determine network stability in the condensed phase of a biological condensate. We hypothesized that a phase-separated system driven by multivalent and strong inter-molecular interactions, such as the PSD condensates studied here, would likely form highly stable and larger molecular network in the condensed phase. Accordingly, molecules bound to the network might be considered as immobile or confined by the network. In contrast, molecular networks in the condensed phase formed by weak but also multivalent molecular interactions (e.g., intrinsic disordered region [IDR]-mediated phase separations) would be more dynamic and the network would also more transient (see *Figure 4A* for a scheme). One might envision that the molecular networks formed in the condensed phase could locally break or reform as molecules within the network could still undergo binding and unbinding processes. Thus, the molecular networks formed in condensed phases are dynamic and heterogeneous. The fraction of time that a molecule stays on the network is correlated with its binding affinity (i.e., the off-rate of the molecule from the network, a value directly related to the dissociation constant of the binding) and avidity (a combination of the available binding sites in the vicinity of the molecule, a parameter related to valency of the molecular interactions in the system and the binding affinity of each individual binding site) between the molecule and the network.

To provide direct experimental support for this general hypothesis, we created a one-component phase separation system, a chimeric protein composed of the prion-like domain (PrLD) of FUS connected with the SAM domain of Shank3 (PrLD-SAMWT, *Figure 4B*). The PrLD of FUS is a well-characterized IDR protein capable of phase separation by itself (*Kato et al., 2012*). The SAM domain of Shank3 can specifically interact with each other in a head-to-tail manner forming large polymers (*Baron et al., 2006*) (see also *Figure 4—figure supplement 1*). Thus, the PrLD-SAM chimera contains a weak and multivalent interaction domain and a specific and multivalent interaction domain within one protein. The PrLD-SAM chimera could undergo phase separation at very low concentrations (see below), so we 'caged' the chimera with highly soluble maltose binding protein (MBP) at its N-terminus and a small, highly soluble protein GB1 at its C-terminus (*Figure 4B*). The resulting 'caged' protein could be purified and concentrated to as high as 500 μM. Cleavage of the caging tags of caged PrLD-SAM with HRV-3C protease induced phase separation of the protein. Substitution of Met1718 of the Shank3 SAM domain with Glu dramatically weakens the head-to-tail interaction of the domain and the mutant SAM domain (SAMME) has very weak propensity of forming oligomers in solution (*Baron et al., 2006*). We also created a 'caged' PrLD-SAMME chimera to investigate the impact of weakening specific interaction on the molecular network formation in the condensed phase (*Figure 4B*). Lastly, we used the FUS PrLD only to investigate the property of the molecular network in the condensed phase that is solely formed by an IDR sequence with only weak multivalent interactions (*Figure 4B*). Again, we 'caged' FUS PrLD at both of its termini with GB1, so that the caged PrLD can be purified in its native form and concentrated to very high concentrations without phase separation.

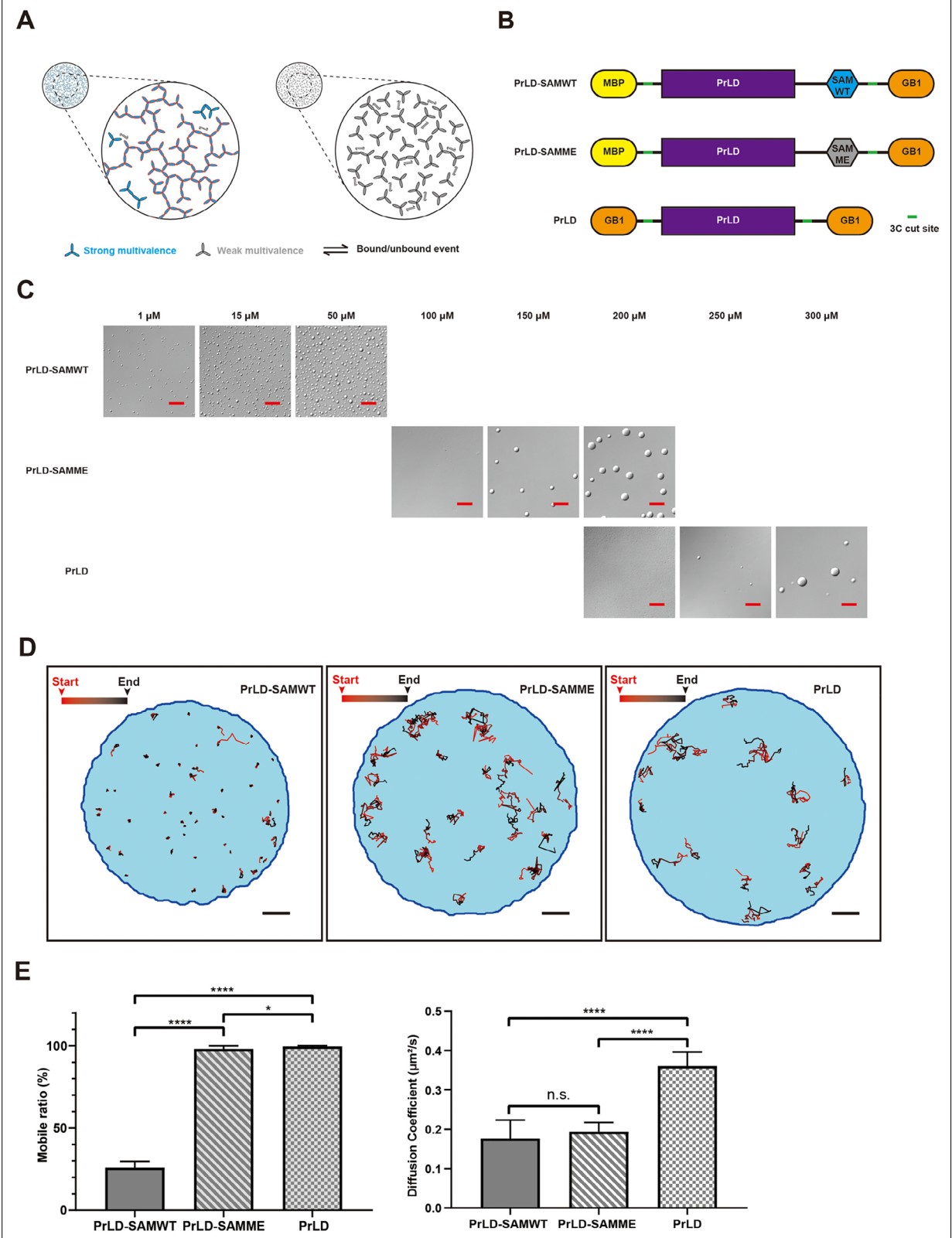

**Figure 4.** Immobilization of molecules by the large dynamic molecular network in the condensed phase of phase-separated systems. (**A**) Schematic illustrations of the concept of stable molecular networks in the condensed phase formed by strongly favorable specific and multivalent interactions (left, blue) versus dynamic molecular networks in the condensed phase formed by relatively weak multivalent interactions (right, gray). Red edge highlights a large dynamic (percolated) network. (**B**) Schematics of the composition of three designed and 'caged' single protein phase separation

*Figure 4 continued on next page*

*Figure 4 continued*

systems with different interaction properties. PrLD, prion-like domain of FUS; SAMWT, WT SAM domain from Shank3; SAMME, the M1718E mutant of Shank3 SAM domain; MBP, maltose binding protein as a caging tag; GB1, the B1 domain of *Streptococcal* protein G as another caging tag. The HRV-3C cleavage sites ('3C cut site') of the proteins are also indicated. (**C**) DIC images showing phase separations of the three designed proteins at different concentrations after removal of the caging tags by HRV-3C protease cleavage. Scale bar: 20 µm. (**D**) Representative tracks showing different motion properties of the three designed proteins in condensed phase. Scale bar: 2 µm. Our analysis of diffusion data from the experiments depicted in this figure was based on 2D projections of 3D diffusion tracks. (**E**) Comparison of Hidden Markov Model (HMM)-estimated mobile ratio in condensed phase (left) and diffusion coefficient in mobile state (right) for the three designed proteins. N = 12, batches of sample with the same condition were used, data are expressed as mean ± standard deviation (SD) with ****p<0.0001, *p<0.0332 by *t*-test. 'n.s.', no significant.

The online version of this article includes the following figure supplement(s) for figure 4:

**Figure supplement 1.** Binding between FUS prion-like domain (PrLD) is extremely weak.

**Figure supplement 2.** Purified FUS prion-like domain (PrLD) takes more than 12 hr to form condensates.

We further demonstrated that the binding between FUS PrLD is indeed extremely weak (*Figure 4—figure supplement 1*). Based on our conceptual framework discussed above, we predicted that the molecular networks in the condensed phase formed by PrLD-SAMWT would be the largest and most stable followed by PrLD-SAMME, and the molecular network of the PrLD condensed phase should be the most dynamic.

We first compared the threshold concentrations of the three proteins for phase separation to occur. Phase separation of each protein was induced by mixing 1 µM HRV-3C protease and immediately injecting the digestion mixture into a sealed, home-made chamber incubated at 20°C. All caged proteins were completely digested within ~30 min after addition of HRV-3C protease. However, the phase separation of PrLD or the PrLD-SAMME chimera took up to 12 hr to occur (i.e., with very slow nucleation rates) (*Figure 4—figure supplement 2*). Thus, we compared DIC images of the three cage-cleaved proteins captured at 12 hr after addition of HRV-3C protease (*Figure 4C*). The PrLD-SAMWT chimera underwent phase separation at concentration as low as 1 µM. In contrast, the threshold phase separation concentrations for PrLD-SAMME and PrLD were much higher (~150 µM and ~250 µM, respectively). These results demonstrate that specific and multivalent interactions act in concert in the PrLD-SAMWT condensate, and such multivalent interactions can dramatically lower the threshold concentration of phase separation (*Espinosa et al., 2020*; *Lin et al., 2022*; *Riback et al., 2020*; *Zeng et al., 2018*; *Zeng et al., 2016*).

We then compared the motion properties of the three proteins in the condensed phase by applying the adaptive single-molecule tracking and HMM method developed above. A concentration some-what higher than the phase separation threshold was used for each protein (i.e., 50, 200, and 300 µM for PrLD-SAMWT, PrLD-SAMME, and PrLD, respectively). Each protein was very sparsely labeled with Alexa 555 (0.02% for PrLD-SAMWT, 0.005% for PrLD-SAMME, and 0.005% for PrLD) to obtain sparse but long lifetime (>1 s) single-molecule tracks. It is clear that in the condensed phase the motion properties of PrLD-SAMWT are dramatically different from those of PrLD-SAMME and PrLD (*Figure 4D*). HMM analyses of experimental data indicate that PrLD-SAMWT molecule spent most of their time (74.0 ± 3.7%) in the transiently confined state, though these molecules were able to switch between the confined and mobile states (*Figure 4E*). In contrast, the mobilities of PrLD-SAMME or PrLD in the condensed phase were much higher. PrLD-SAMME spent 98.1 ± 1.8% of time in the mobile state and PrLD spent 99.6 ± 0.4% in the mobile state (*Figure 4E*). These findings indicated that PrLD-SAMWT molecules in the condensed phase likely formed a very large but still dynamic molecular network due to the presence of specific and multivalent SAM-SAM interaction. In contrast, the molecular networks in the condensed phase formed solely by weak and multivalent interactions were probably much smaller and much more dynamic. Interestingly, the HMM-estimated diffusion coefficients of PrLD-SAMWT and PrLD-SAMME in their mobile state were very similar (0.18 ± 0.05 µm$^2$/s vs. 0.19 ± 0.02 µm$^2$/s) (*Figure 4E*), indicating that PrLD-SAMWT and PrLD-SAMME have similar effective molecular sizes in their mobile state. The mobile state of both proteins likely corresponds to each molecule not bound to the large, essentially immobile molecular networks in the condensed phase. The diffusion coefficient of the mobile state of PrLD is estimated by HMM to be 0.36 ± 0.04 µm$^2$/s (*Figure 4E*) and the molecular weight of PrLD is about half of PrLD-SAMWT and PrLD-SAMME, again suggesting that the mobile state of PrLD likely corresponds to the network-unbound form of PrLD. Taken together, the above single-molecule tracking study of our designed FUS PrLD systems buttressed our general

hypothesis that strong multivalent interactions tend to lead to formation of large and more stable molecular networks in the condensed phase, which could dramatically reduce the overall motions of the molecules in the condensed phase. Most prominently, proteins that bind to these large and stable molecular networks no longer conform to free diffusion processes found in dilute solutions. Instead, in such condensed phases, proteins switch between immobile/confined states and more freely diffusive states, corresponding respectively to the network-bound and relatively free forms. By comparison, in condensed phases formed by weakly interacting IDR sequences, the molecular networks in the condensed phase are much more dynamic/transient and likely much smaller in spatial extent, thus resulting in IDR proteins displaying simple diffusion-like behaviors.

## Fluorescence recovery after photo-bleaching (FRAP) in phase separation systems

FRAP assays are widely used to examine dynamic properties of phase separation systems. Quantitative theory for analyzing FRAP results in phase separation systems based on Flory–Huggins theory have been developed for weak interaction systems (*Hubatsch et al., 2021*). For heterogeneous condensed phase systems, traditional FRAP experiment might not be a good way to test the liquid-like properties (*McSwiggen et al., 2019*). As we have shown above, motion properties of molecules in the condensed phase are radically different for systems involving strong and specific multivalent interactions compared to the system with only weak interactions. Seeking a better understanding of the general trend, we simulated FRAP properties of several different model phase separation systems pertinent to our experiments using representative diffusion coefficients (provided in the legend for *Figure 5*) in a manner similar to the test runs reported in *Table 1*. Following this investigative logic, a 2D phase separation system mimicking phase separation on a flat surface such as lipid membranes was constructed with three different sizes of circular-shaped condensed phase (radius of 0.5, 1, and 2 μm) in a box with periodic boundary conditions (*Figure 5A*). By monitoring the simulated molecular positions for 100 s after photo-bleaching in this model, we can simulate the FRAP curve of any region in the box. We further define a region with certain size to be bleached as region of interest (ROI).

Using this simple model construct, we first asked whether the size of ROI vs. the size of condensed phase may affect FRAP results. To address this question by simulation, the diffusion coefficients of the molecules were set to 0.01 μm²/s and 1 μm²/s, respectively, for the condensed and dilute phases. The mobile ratio was set at 100% (i.e., simulating FRAP curves for a phase separation system dominated by weak multivalent interactions). This setting leads to 100 times enrichment of the molecule into the model condensed phase. The ROIs with a radius of 0.5 μm were selected at the center of the three different sized droplets (ROI 1, 2, 3 in *Figure 5A*). The simulated results showed that the apparent recovery curve for ROI 3 was considerably slower than the first two ROIs (*Figure 5B*), as all molecules in ROI 3 needed to diffuse from the dilute phase into the entirely bleached condensed phase. This simulation indicates that one should select condensed phases with similar sizes when comparing FRAP curves of related phase separation systems. When possible, always select a small ROI within a large droplet for FRAP analysis.

We next simulated FRAP curves for the model phase separation systems containing specific interactions. We first set the mobile ratio of simulated molecules in condensed phase at 10%, with average lifetime of the confined state being 10 s. To maintain the same apparent diffusion coefficient and the same enrichment level in the condensed phase as those in the system with no molecular confinement described above, the diffusion coefficient of the molecule in the mobile state within the condensed phase was chosen to be 0.1 μm²/s. We simulated the FRAP curves for the same three ROIs as indicated in *Figure 5A*. Although the FRAP recovery speed for ROI 3 was still slower than the other two ROIs, the difference became smaller (*Figure 5C*). Further increasing the lifetime of the confined state to 100 s while keeping other parameters unchanged led to linear-like slow recovery curves and the differences among the three ROIs were further diminished (*Figure 5D*). In an extreme case where the lifetime of the confined molecules was infinitely long (i.e., extremely strong binding resulting in behaviors similar to those of unrecoverable aggregates), all three recovery curves looked similar with a fast recovery speed toward their maximal recovery level at 10% (*Figure 5E*). These simulations suggest that the property of the molecular network in the condensed phase can have dramatic impact on its FRAP recovery rate in the system. For example, a slow recovery speed from a FRAP experiment does not necessarily mean that all of the molecules in condensed phase move slowly. Instead, it may

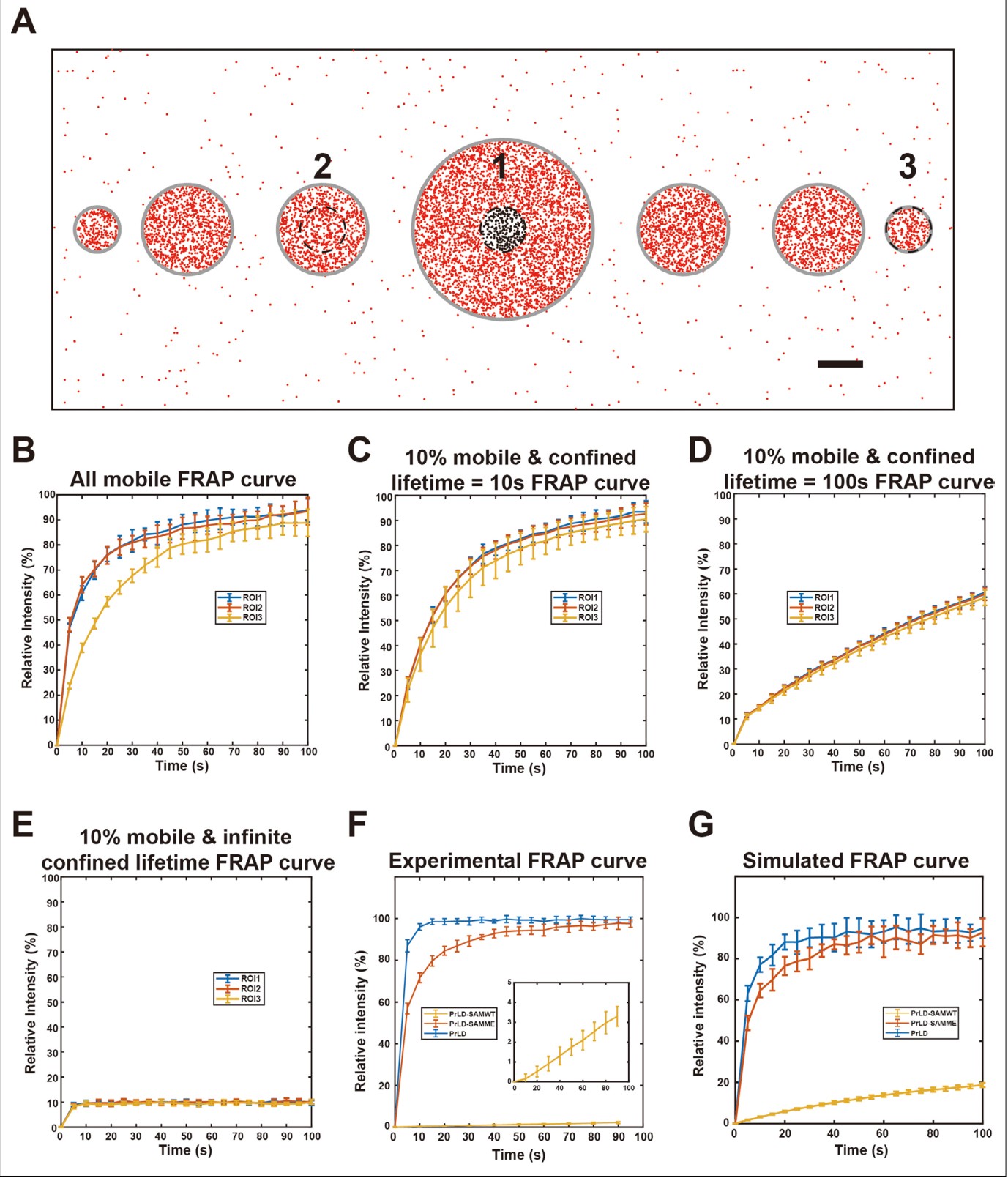

**Figure 5.** Simulated and experimental fluorescence recovery after photo-bleaching (FRAP) curves of the FUS prion-like domain (PrLD) systems. (**A**) Schematic representation of the phase separation system for the present FRAP simulations. Simulations were conducted in a 20 μm × 8 μm box with periodic boundary conditions. Three regions of interest (ROIs) (1, 2, 3) with a fixed diameter of 0.5 μm are positioned at the center of three different sized droplets (2, 1, and 0.5 μm in diameters, as indicated) were selected for photo-bleaching. Gray lines indicate the phase boundaries of the droplets.

*Figure 5 continued on next page*

*Figure 5 continued*

Black dots represent bleached molecules that can exchange with unbleached molecules in red. scale bar: 1 µm. (**B**) Simulated FRAP curves of the three ROIs when all molecules in the condensed phase are mobile, with an enrichment fold of 100 and diffusion coefficients in the condensed and dilute phases being 0.01 µm²/s and 1 µm²/s, respectively. Data are expressed as mean ± standard deviation (SD) from 10 independent simulations. (**C**) Simulated FRAP curves of the three ROIs when only 10% of molecules in condensed phase are mobile and diffusion coefficients in the condensed and dilute phases being 0.1 µm²/s and 1 µm²/s, respectively. The lifetime of the molecule in the confined state was set at 10 s. (**D**) Same as in (**C**) except that the lifetime of the molecules in the confined state was set at 100 s. (**E**) Same as in (**C**) except that the molecules in the confined state were treated as permanently immobilized. (**F**) Experimental FRAP curves of PrLD, PrLD-SAMME, and PrLD-SAMWT condensates. In each case, a photo-bleaching region with the size of 1.95 µm in diameter was selected inside a large droplet (see ***Figure 5—figure supplement 1***). The zoom-in panel is an expanded view of the FRAP curve of PrLD-SAMWT. Data are expressed as mean ± SD, with recovery experiments performed on 10 different droplets. (**G**) Simulated FRAP curves of the three designed proteins in the condensed phase using the parameters derived from the experiments described in ***Figure 4*** (see 'Materials and methods' for quantitative details). The region selected for photo-bleaching has a diameter of 2 µm and is located in a model condensed phase of infinite size. Data are expressed as mean ± SD from 10 independent simulations.

The online version of this article includes the following figure supplement(s) for figure 5:

**Figure supplement 1.** Representative confocal images showing fluorescence recovery after photo-bleaching (FRAP) experiments of the condensed droplets formed by prion-like domain (PrLD), PrLD-SAMME, or PrLD-SAMWT.

mean that a fraction of molecules spend most of the time bound to an essentially immobile molecular network with very slow dynamics. Once switched into the mobile state, molecules can diffuse quite freely and rapidly.

The above physical picture revealed by our simulation data are supported by experimentally measured FRAP curves of the three-phase separation systems shown in ***Figure 4B***. A small ROI with identical radius within a relatively large, condensed droplet was selected for the FRAP experiments in each of the three systems (***Figure 5—figure supplement 1***). The PrLD system showed the fastest recovery speed, and the PrLD-SAMME system showed a slightly slower recover speed. The PrLD-SAMWT system, with its large and stable molecular network in the condensed phase, displayed a very slow and near linear recovery curve (***Figure 5F***). In addition to the simulations of model systems in ***Figure 5B–E*** with diffusion coefficients and mobile ratios selected a priori, we further simulated the FRAP curves of the three systems using the diffusion coefficients and confinement ratios derived experimentally from the single-molecule tracking data (i.e., the parameters shown in ***Figure 4E***; see 'Materials and methods' for details). The simulated FRAP curves were overall very similar to those obtained experimentally (***Figure 5G vs. F***), notwithstanding a faster recovery rate in the simulated curve vs. the experimental curve for the PrLD-SAMWT system that is likely due to overestimations of the mobile ratio of the protein in the tracking experiment. Taken together, the above theoretical and experimental studies revealed that the FRAP curve of a molecular species in a phase separation system is heavily influenced by the proportion of time of the molecules that are transiently trapped in the confined state, a parameter directly linked to the strong and specific multivalent interactions of the system. Accordingly, a low recovery rate in the FRAP assay does not necessarily mean that a large fraction of molecules in the condensed phase is permanently immobile.

## Discussion

In this study, we developed a method that can track single-molecule motions in both condensed phase and dilute phase simultaneously by using photo-switchable dye-labeled proteins. To accommodate the heterogeneity of both the distributions and diffusion modes of molecules in the condensed and dilute phase, an adaptive single-molecule tracking algorithm was developed by setting the optimized search range for molecules with different diffusion coefficients. The method is simple and highly robust. It can be deployed to track single-molecule motions of phase separation systems with very broad dynamic ranges including highly dynamic systems formed by intrinsically disordered proteins or very stable (or highly percolated) systems formed by strong and likely structure-specific multivalent molecular assemblies such as PSDs. With implementations of sparse labeling techniques such as the HaloTag labeling system or photo-convertible fluorescent proteins like mEos (***Inavalli et al., 2019***; ***Los et al., 2008***), our method can be applied to track single-molecule motions of biological condensates in living cells.

Applying this technique to PSD, the most notable direct experimental finding from our study is that molecules in the condensed phase of biological condensates do not exhibit simple diffusion

behaviors as those observed in dilute solutions. Instead, molecules appear to constantly switch between a broadly construed transient confined state and a broadly construed mobile state in the condensed phase. The general features of this phenomenon are clear from our raw tracking data. Here, semi-quantitative accounts of this discovery are provided by an HMM as well as an analysis that does not presume a particular model of diffusion and is based only upon the correlation of consecutive displacements. This two-state-like phenomenon of condensed-phase motion is most likely underpinned by phase separation-mediated formation of large and percolated molecular networks. The size and dynamic properties of the molecular network in the condensed phase are necessarily determined by the binding affinity (or affinities) and valence of the interaction(s) of the molecule(s) in the phase separation system, a trend that has been recently predicted by theoretical simulations treating biomolecules as associative polymers (*Choi et al., 2020*; *Harmon et al., 2017*; *Pappu et al., 2023*) and illustrated by the model PrLD-SAMWT system examined here. The fraction of time and the time duration that a molecule spends in the confined state is also determined by the binding affinity of the molecule to, the complexity, and the dynamic properties of the network. For biological condensates assembled largely by strong and specific molecular interactions (including condensates formed with combinations of strong molecular interactions with IDR sequence-based weak interactions; *Chen et al., 2020*; *Feng et al., 2021*; see *Lin et al., 2022* for more discussions), molecules in the condensed phase spend most of their time in the confined state. Since the fraction of time and the time duration that a molecule spends in the confined state vs. those in the mobile state are basic defining parameters for the functions of molecules in any reaction systems (e.g., binding/unbinding rates, kinetics of enzyme catalysis, lifetime/dwelling time a molecule in a molecular machinery, etc.), our finding reveals a fundamental aspect of molecular properties created by biological condensates that is distinctly different from that in dilute solutions. In this regard, our study implies further that, in principle, virtually unlimited types of biological condensates with very broad dynamic network properties may form by utilizing existing repertoires of proteins and nucleic acids via different combinations of binding affinities and interaction valences. In this light, phase separation-mediated formation of biological condensates is further revealed to be a very powerful means indeed for cells to form numerous subcellular organelles with a continuum of dynamic and material properties ranging from very dynamic, dilute solution-like assemblies to highly stable, solid-like systems. Confinements of molecules in cellular condensates have been observed by single-molecule tracking experiments recently (*Chong et al., 2022*; *Miné-Hattab et al., 2021*; *Niewidok et al., 2018*). However, since the molecular compositions of cellular condensates cannot be easily defined, the mechanistic bases underlying the confined state of molecules in these cellular condensates are difficult to be discerned. Single-molecule tracking experiments using the reconstituted and compositionally defined phase separation systems in the current study allowed delineation of the mechanism underlying the unique motion properties of molecules in the condensed state.

We have demonstrated that a two-motion-state model can provide a semi-quantitative rationalization of our experimental data and, at the same time, offer a conceptual framework for their physical interpretation. Indeed, such a two-motion-state picture is well motivated by direct experimental observations (*Figure 2E*, *Video 1*). Nonetheless, given the inherent complexity of biological condensates, it is intuitive to recognize that molecules in the condensed phase corresponding to a biological condensate would most likely have multiple motion states due to the heterogeneity of intra-condensate interactions (cf. schematics in *Figure 3G*). With this in mind, the two motion states in our simple two-state model for condensed-phase dynamics should be understood to be consisting of multiple substates. For instance, one might envision that the percolated molecular network in the condensed phase is not uniform (e.g., existence of locally denser or looser local networks) and dynamic (i.e., local network breaking and forming). Therefore, individual proteins binding to different subregions of the network will have different motion properties/states. It follows that molecular motions in the percolated condensed phase likely cover a continuum of mobilities, although the confined- and mobile-state diffusion coefficients deduced from our experimental measurements based on the two-state model do represent two important time scales of intra-condensate dynamics of PSD. In light of this basic understanding, the 'confined state' and 'mobile state' as well as the derived diffusion coefficients in this work should be understood as reflections of ensemble-averaged properties arising from such an underlying continuum of mobilities. Further development of experimental techniques in conjunction with more refined models of anomalous diffusion (*Joo et al., 2020*; *Kuhn et al., 2021*;

*Muñoz-Gil et al., 2021*) will be necessary to characterize these more subtle dynamic properties and to ascertain their physical origins. It will also be interesting to explore a possible relationship between our multiple-mobility model and the phenomenon of motility-induced phase separation (*Cates and Tailleur, 2015*).

Since the interactions between molecules can be modulated via numerous cellular processes such as posttranslational modifications, protein biogenesis/turnovers, epigenetic modification, cellular milieu alterations, etc., the dynamic network properties and consequently the functions of organelles formed via phase separation may be regulated in ways that are distinctly different from those occurring in dilute solutions. Compared to the rich knowledge and quantitative theories for dilute-solution systems, few satisfying theoretical frameworks have been established for the condensed assemblies formed via phase separation in cells (see *Mittag and Pappu, 2022* and refs therein), partly due to our poor understandings of microscopic motion properties of molecules in the condensed phase. The dramatic dynamic and material property differences of condensates formed by weakly associative intrinsically disordered proteins and by biomolecules with strongly favorable specific interactions involving folded domains indicate that biological phase separation research that focuses primarily on IDRs has only touched the tip of the iceberg. Much knowledge awaits discovery by placing more emphasis on the diverse roles of strong, structure-specific interactions and their synergy with relatively weaker IDR interactions in the assembly, dynamics, and ultimately the physiological functions of biological condensates.

## Materials and methods
### Protein expression and purification
Constructs for expression of Trx-His-GCN4-NR2B, PSD-95 (UniProt: P78352-1), Shank3 (UniProt: Q4ACU6), GKAP (UniProt: Q4ACU6-1), and Homer3 (UniProt: Q9NSC5-1) were described previously (*Zeng et al., 2018*). MBP-His$_8$-GCN4-NR2B tetramer was created by inserting GCN4-NR2B sequence into an in-house modified pET32a vector. MBP-His$_8$-GCN4-NR2B trimer and dimer were mutated from the tetramer version by changing the hydrophobic residues in the GCN4 domain (*DeLano and Brünger, 1994*). Constructs of GB1-PrLD-GB1 contained FUS-PrLD (UniPort: P56959, segment: 1–212) with the protecting GB1 protein fused to the N- and C-terminal ends. MBP-PrLD-SAMWT-GB1 is a fusion protein with the SAM domain of Shank3 (aa 1654–1730) fused to the C-terminal end of PrLD, and the resulting chimeric protein was further protected by tagging its N-terminus with MBP and C-terminus with GB1. MBP-PrLD-SAMME-GB1 is the same as MBP-PrLD-SAMWT except that Met1718 in the SAM domain was replaced by Glu. An additional cysteine was inserted at the N-terminus of PrLD for cysteine labeling. All constructs were confirmed by DNA sequencing. Recombinant proteins were expressed in *Escherichia coli* BL21 (DE3) cells in LB medium at 16°C. Protein expressions were induced by adding 0.25 mM IPTG when OD$_{600}$ reached 0.6–0.8. His$_8$-tag containing recombinant proteins were purified using Ni$^{2+}$-NTA agarose affinity column followed by size-exclusion chromatography (Superdex 200 26/60 column from GE healthcare) in a final buffer containing 100 mM NaCl, 50 mM Tris-HCl (pH 7.8), 1 mM DTT, and 1 mM EDTA. The purified proteins (except the NR2B proteins for lipid binding and PrLD/PrLD-SAMME/PrLD-SAMWT) were then subject to tag removal by HRV-3C or TEV protease at 4°C overnight followed by another round of size-exclusion chromatography. All purified proteins were checked to ensure that they are free of nucleic acid contamination.

### Protein fluorescence labeling
His$_8$-tagged NR2B proteins were labeled with Alexa 647 NHS ester (Thermo Fisher) and PrLD-SAMWT/PrLD-SAMME/PrLD proteins were labeled with Alexa 555 maleimide (Thermo Fisher). Alexa 647 NHS ester was first dissolved in DMSO at a concentration of 10 mg/mL. Before labeling, all purified proteins were exchanged into a Tris-free buffer containing 100 mM NaHCO$_3$ (pH 8.4), 100 mM NaCl, and 1 mM EDTA (plus 1 mM DTT for NR2B) using a HiTrap desalting column. NR2B was concentrated to 20–50 µM and mixed with the corresponding dye at a 1:1 molar ratio. Alexa 555 maleimide was dissolved in DMSO at a concentration of 10 mg/mL and mixed with PrLD-SAMWT/PrLD-SAMME/PrLD (>100 µM, without DTT) at a 1:1 molar ratio. The mixture was incubated at room temperature for about 1 hr, and the reaction was terminated by adding 200 mM Tris-HCl (pH 8.2). The mixture was next loaded to a HiTrap desalting column to separate the unreacted fluorophores and exchange proteins

into buffers for following experiments. Efficiency of individual labeling was measured by Nanodrop 2000 (Thermo Fisher). Unlabeled protein was mixed with each labeled protein to adjust the final labeling ratio needed for imaging experiments.

## Fast protein liquid chromatography coupled with static light scattering (FPLC-SLS) assay

The analysis was performed on an AKTA FPLC system (GE Healthcare) coupled with a static light scattering detector (miniDawn, Wyatt) and a differential refractive index detector (Optilab, Wyatt). Protein samples (concentrations for each reaction were indicated in the figure legends) were filtered and loaded into a Superose 12 10/300 GL column pre-equilibrated by a column buffer composed of 50 mM Tris, pH 8.2, 100 mM NaCl, 1 mM EDTA, and 2 mM DTT. Data were analyzed with ASTRA6 (Wyatt).

## Lipid preparation

POPC (Avanti Lipids, Cat# 850457P), DGS-NTA-Ni$^{2+}$ (Avanti Lipids, Cat# 790404P), and PEG-5000 PE (Avanti Lipids, Cat# 880230P) were first solubilized in chloroform to a stock concentration of 20 mg/mL, 10 mg/mL, and 1 mg/mL. Lipid mixture containing 98% POPC, 2% DGS-NTA-Ni$^{2+}$, and 0.1% PEG-5000 PE was dried under a stream of nitrogen gas followed by vacuum pumping to evaporate chloroform thoroughly. The dried lipids were then resuspended in PBS to a final concentration of 0.5 mg/mL. Multilamellar vesicle solution was next solubilized by 1% w/v sodium cholate and loaded onto a desalting column. During the desalting process, sodium cholate was diluted to allow small unilamellar vesicles (SUVs) to form in the buffer containing 100 mM NaCl, 50 mM Tris-HCl (pH 7.8), and 1 mM TCEP (the 2D buffer).

## Coating chambered cover glass with lipids

Chambered cover glass (Lab-tek) was immersed in Hellmanex II (Hëlma Analytics) overnight. Following extensive rinsing with MilliQ H$_2$O, the chambered cover glass was then washed with 5 M NaOH for 1 hr at 50°C and then thoroughly rinsed with MilliQ H$_2$O to remove NaOH. The cleaned coverslips were washed three times with the coating buffer (50 mM Tris, pH 8.2, 100 mM NaCl, 1 mM TCEP). Typically, 150 μL SUVs were added to a cleaned chamber and incubated for 1 hr at 42°C, resulting in the SUVs fully collapsing on glass and fusing to form SLBs. Chambers with SLBs were then gently washed three times each with 750 μL of coating buffer to remove extra SUVs before being blocked by the clustering buffer (coating buffer plus 1 mg/mL of BSA) for 30 min at room temperature.

## Phase separation on SLB

The SLBs contained 2% DGS lipid with Ni$^{2+}$-NTA attached to its head. We used GCN4-NR2B with an N-terminal thioredoxin (TRX)-His$_8$ tag (referred to as NR2B tetramer) to attach to SLBs via binding to DGS-NTA-Ni$^{2+}$. The NR2B (4 μM final concentration) tetramer was added to an SLB-containing chamber. After 30 min of incubation at room temperature, the chamber was washed with the clustering buffer for three times (each time at 750 μL volume) to remove excessive NR2B tetramers. PSD-95, Shank3, GKAP, and Homer3 (each at 2 μM final concentration) were sequentially added into the system. Imaging acquisition started at 15 min after adding all components.

## Phase separation in 3D solution in chamber

For PSD condensates, 10 μM of five PSD proteins were mixed and injected into a homemade chamber and sealed immediately (*Zeng et al., 2016*). The mixtures were incubated for 15 min before starting image acquisitions. For the PrLD/PrLD-SAMME/PrLD-SAMWT systems, 300/200/50 μM of 'caging' tag-containing protein was mixed with 1 μM of HRV-3C protease, each mixture was injected into a homemade chamber and sealed immediately. The samples were incubated at 20°C for 12 hr before image acquisitions.

## dSTORM imaging

Freshly prepared imaging buffer (the 2D buffer plus 1% D-glucose [Sigma G8270], 5.6 μg/mL glucose oxidase [Sigma G2133-50KU], from 100× stock prepared in the coating buffer), 40 μg/mL catalase (Sigma C9322-10G, from 100× stock prepared in the coating buffer), and 15 mM β-mercaptoethanol

were injected into an imaging glass chamber to replace the original coating buffer. Imaging of each sample was completed within 30 min upon addition of the imaging buffer.

dSTORM images for the condensates formed on SLBs were taken by a home-built two-color super-resolution localization microscope based on a Nikon Ti-E inverted microscope body (*Zhao et al., 2015*). Here only one channel was used to image samples labeled with Alexa 647. A ×100 objective lens (CFI Apo TIRFM ×100 Oil, N.A. 1.49, Nikon) was used to observe the fluorescence signals. An EMCCD (electron-multiplying charge-coupled device, Andor, IXon-Ultra) was applied to collect the emission lights that passed through a channel splitter. For each sample, 2000 frames of images with an exposure time of 30 ms/frame were captured from at least six different areas. The laser intensity was fixed at 1 kW/cm$^2$ during the imaging, and the microscope was at the TIRF mode. If single-molecule signal density of a sample was too high, a pre-image photo-bleaching with a strong laser intensity (4.0 kW/cm$^2$) was used to reduce the single-molecule signal density. The TIRF raw images were processed by Rohdea (Nanobioimaging Ltd, Hong Kong) to generate the localization coordinates in each frame.

dSTORM images for 3D phase separation system were taken with a Zeiss Elyra7 microscope with a ×63 oil objective lens. Samples were first bleached with a full power laser (500 mW) and then imaged with 20% of the full power of the 488/561/641 nm lasers with the HILO mode illumination. A TIRF-hp filter was used during imaging. For each sample, 4000 images were captured using an exposure time of 30 ms/frame. Autofocus with the 'definite focus' strategy was performed at every 500 frames. Maximum point spread function size was set at 9 and signal-to-noise ratio was set at 5 when capturing single molecules with Zeiss Elyra7. Samples were labeled with 0.005–1% ratio of dyes depending on the signal density.

## Adaptive single-molecule tracking algorithm

The heterogeneity of molecule distributions and diffusion modes in the dilute and condensed phases of liquid–liquid phase separation systems requires an adaptive single-molecule tracking algorithm to minimize the track assign error locally and globally. Traditional single-molecule tracking in high-density systems (*Jaqaman et al., 2008*; *Manley et al., 2008*; *Tinevez et al., 2017*) usually set a global search range (step limit) manually to connect molecular tracks. A step limit that is too small will cause lots of missing connection for molecules with fast diffusions, whereas a step limit that is too large will cause lots of false positive connections for molecules with slow diffusions. Such errors cannot be fixed in post-tracking data analysis. A biological condensate system typically contains a condensed phase with slow-diffusing molecules coexisting with a dilute phase with fast-diffusing molecules, and molecules in the condensed phase constantly switch between mobile state and confined state. A single global maximum step limit without any prior knowledge might not be suitable for tracking molecules in a phase separation system. A typical solution for motion switch is to use the HMM to fit a diffusion model that contains diffusion coefficients (*D*) and switching probabilities (*P*) for different diffusion states (*S*) (*Das et al., 2009*; *Persson et al., 2013*). Taking all of the above factors into consideration, we developed a new algorithm by adaptively choosing potentially different maximum step limits for different diffusion states to link the localizations into tracks and using HMM to fit a two-state diffusion model in the condensed phase.

The track assignment errors can be divided into two parts, true negatives (TN) and false positives (FP). A true negative error is defined as an existing track was not linked, which leads to misses of long-distance steps. This part of the error can be estimated by a Gaussian distribution if we assume that molecules essentially undergo Brownian motion in the mobile state, that is, the true-negative (TN) error (*E*) is given by $E_{TN} = e^{-\frac{R^2}{4Dt}}$, where *R* is the search range (step limit). A false positive error is defined as linking of a nonexistent track. This part of the error may be estimated by the usual expression for the collision frequency of molecules with a 1D cross-section corresponding to the linear length scale of a 2D circle with a diameter equal to the search range because this collision frequency is a good estimate of the number of times two different molecules come within a spatial separation of *R* within time interval *t* due to diffusion. It follows that $E_{FP} = \sqrt{\pi Dt} \times \sigma R$, where $\sigma$ is molecule density in 2D. Thus, the estimated assignment error under a certain search range *R* may be quantified as

$$E = E_{TN} + E_{FP} = e^{-\frac{R^2}{4Dt}} + \sqrt{\pi Dt} \times \sigma R \qquad (3)$$

(Note that $R$ may also be used as an alternate 1D collision cross-section instead of the $\sqrt{\pi}R/2$ value used above for the estimation of $E_{FP}$; such a change is immaterial as it leads only to negligible quantitative differences in the results below.) To find the search range $R$ that entails minimal error under certain $\sigma$ and diffusion coefficient $D$, we solve the extremum condition $\frac{dE}{dR} = 0$ with $\frac{d^2E}{dR^2} > 0$ using the expression for $E$ in Equation (3). Since the root mean square displacement (RMSD) in time interval $t$ is given by $\sqrt{4Dt}$ , we may define a dimensionless quantity $X = \frac{R}{RMSD} = \frac{R}{\sqrt{4Dt}}$ as the ratio of search range to the RMSD for time $t$ of any given molecule. The equation for the first derivative $\frac{dE}{dR} = 0$ can then be rewritten as $Xe^{-X^2} = \sqrt{\pi}\sigma Dt$, and it is straightforward to verify that the second derivative $\frac{d^2E}{dR^2} = \frac{e^{-x^2}}{2Dt}\left(2X^2 - 1\right) > 0$ if $X > 1/\sqrt{2}$ . The fluorophore density in the condensed phase and the dilute phase were typically $\sigma = 0.20$—$0.40/\mu m^2$ and $0.01$—$0.02/\mu m^2$, respectively; diffusion coefficients were $D = 0.02$—$0.2$ $\mu m^2/s$ and $0.5$—$2.0$ $\mu m^2/s$, respectively. Thus, with a per-frame time $t = 30\text{ms}$ , the corresponding variation in the value of $\sqrt{\pi}\sigma Dt$ is between $\approx 2.7 \times 10^{-4}$ and $4.3 \times 10^{-3}$ . The solution of $X$ for the first-derivative condition under such conditions was within a very small region of 2.5—3.0 (*Figure 2—figure supplement 1A*), thus satisfying the second-derivative condition of $X > 1/\sqrt{2}$ for minimal error as well. We could roughly estimate the diffusion coefficients by displacement distributions (*Figure 3A and B*; *Hansen et al., 2018*) starting with a default search range (500 nm). We then moved on to find an optimized search range and use this optimized search range to complete the final track assignments (illustrated in *Figure 2A*).

We have verified the above theoretical formulation for obtaining an optimized $R$ (search range) by simulated molecular systems undergoing homogeneous Brownian motions with different diffusion coefficients and densities (details for these simulations are described under different subheadings below). For systems that are similar to our 2D PSD system, simulation showed the same converged optimal $X$ at ~2.5 (*Figure 2—figure supplement 1B–E*). We simulated many other conditions with different molecular densities ($N$) and diffusion speeds (RMSD) in both dilute and condensed phases, and all showed a similar optimal $X$ of ~2.5 (*Figure 2—figure supplements 2 and 3*). For systems with molecules undergoing switching between confined state and mobile state in the condensed phase (defined as fraction of mobile state or mobile ratio, $P_m$) and with different lifetime in the mobile state $t_m$, the optimal $X$ value also converged to ~2.5 (*Figure 2—figure supplement 4*). Taken together, these simulation results indicated that the optimal $X$ value was very similar under different conditions, at least for the diverse set of conditions we have tested. Thus, we used a default $X = 2.5$ to determine the optimized $R$ (search range) for all the experiments in the present study.

To further validate the algorithm experimentally, we prepared $His_6$-tagged MBP fused with a GCN4 dimer, trimer, or tetramer. The three fusion proteins have a molecular weights ratio of 2:3:4 measured by light scattering experiment (*Figure 2—figure supplement 1F*). We measured the diffusion coefficients of the three MBP proteins coated onto SLBs with our adaptive single-molecule tracking algorithm without any preset parameters (*Figure 2—figure supplement 1G*). The measured diffusion coefficients for the MBP-GCN4 dimer, trimer, and tetramer are very close to 1/2:1/3:1/4, which are precisely the expected theoretical ratios for the three proteins on SLB. Thus, this test set of experimental data supports that our developed algorithm is robust in adaptively determining the diffusion coefficients without any prior knowledge.

## Correlation-based classification of high- and low-mobility displacement steps without presuming simple diffusion

Because presumption of a motion type such as a combination of simple Brownian diffusions may artificially introduce an unwarranted separation of time scales and other possible biases, we developed a model-independent approach to analyze our experimentally determined displacements (*Figure 3—figure supplement 3*). To gain quantitative physical insights into NR2B molecular motion within the condensed phase, we consider a total of 2522 experimental trajectories that are configured entirely in the condensed phase (i.e., no transition to the dilute phase) and each consists of at least 10 steps (i.e., 11 positions). The total number of displacements $d_i$ in this set of trajectories is 41,854, with an overall distribution $P(d_i)$ shown by the black curve in *Figure 3—figure supplement 3A*. To probe the extent to which NR2B motion in PSD condensed phase deviates from simple diffusion, we divided the experimental displacement into two classes, namely $d_i < d_b$ and $d_i > d_b$, using a boundary or demarcation $d_b$ between small and large displacements (an optimized value of which is to be determined; note that

there is no $d_i = d_b$ for the $d_b$ values we considered) and then obtained two conditional distributions of displacements $P(d_{i+1}|d_i < d_b)$ and $P(d_{i+1}|d_i > d_b)$ for $d_{i+1}$ given that the preceding displacement $d_i$ is, respectively, less than or larger than $d_b$ (see, e.g., the blue and orange histograms in *Figure 3—figure supplement 3A* for $d_b$ = 60 nm). If overall condensed-phase NR2B motion were a simple diffusion process, the distribution of displacement of a given step should be independent of the displacement of the previous step and thus the two conditional distributions should be identical; but *Figure 3—figure supplement 3A* demonstrates that the two conditional distributions are significantly different, indicating once again that condensed-phase NR2B motion cannot be described as a simple diffusion process.

This observation from *Figure 3—figure supplement 3A* suggests that we may, as a first approximation, attempt to classify condensed-phase NR2B motion into a low-mobility (L) and a high-mobility (H) states by using a demarcation $d_b$ that results in maximal difference between the low- and high-mobility conditional displacement distributions. Here we use the overlap coefficient (denoted OVL) to quantify the L-H difference in conditional displacement distribution. In general, OVL is a measure of how much two distributions overlap, viz., $\mathrm{OVL} = \int dx \min \left[ P\left( x \right), Q\left( x \right) \right]$, where *P*(x) and *Q*(x) are the distributions of interest (i.e., OVL is the area of the overlapping region of the two distributions; see, e.g., *Song et al., 2017*). We computed OVL among the overall displacement distribution, the L and H conditional displacement distributions and found that the OVL between the H and L conditional displacement distributions reaches its minimum (i.e., with maximally difference) at around $d_b$ = 60 nm (*Figure 3— figure supplement 3B*). We therefore adopt $d_b$ = 60 nm as a workable demarcation between low- and high-mobility states of NR2B motion in the model PSD condensate. This classification is illustrated by the example provided in *Figure 3—figure supplement 3C*. In general, among a set of ~40 randomly selected experimental trajectories we inspected in detail, a classification using $d_b$ = 60 nm appears reasonable although in some cases some of the relatively short displacements are slightly longer than 60 nm. We may then proceed to use $d_b$ = 60 nm to resolve the overall displacement distribution into a low-mobility (L) component (*Figure 3—figure supplement 3D1*) and a high-mobility (H) component (*Figure 3—figure supplement 3D2*). As shown in the figure, the L-component distribution may be reasonably fitted (albeit with some deviations, see below) to a simple-diffusion functional form with a parameter *s* = 13.6 ± 3.7 nm, where *s* may be interpreted as a microscope detection error due to imaging limits or alternately expressed as $s = D_L t$ with $D_L$ = 0.006149 μm²/s being the fitted confined-state diffusion coefficient and t = 0.03 s is the time interval of the time step between experimental frames. (The HMM-estimated confined-state $D_c$ = 0.0127 μm²/s corresponds to *s* = 19.5 nm.) If we fit the H-component distribution to a simple diffusion process, the best-fitted diffusion coefficient for the mobile state is $D_m$ = 0.044 ± 0.05 μm²/s (blue curve in *Figure 3—figure supplement 3D2*). Although a Lévy distribution (*Viswanathan et al., 2008*; *Wang et al., 2020*) for displacements in anomalous diffusion (*Bouchaud and Georges, 1990*; *Joo et al., 2020*) offers a clearly superior fit (orange curve in *Figure 3—figure supplement 3D2*) and should be explored for more detailed analysis of the H- as well as L-component displacement distributions in future studies, it suffice for our present purpose to limit our discussion of condensed-phase motion largely to a combination of simple diffusion processes depicted by the fitted blue curves in *Figure 3—figure supplement 3D1 and D2* . In that case, with the confinement ratio ($P_c$ = 86.5%) and mobile ratio ($P_m$ = 13.5%) provided directly by the number of displacements classified as 'L' and 'H' in accordance with a $d_b$ = 60 nm demarcation, the combination of the two fitted simple diffusion processes is seen to provide a reasonably good approximation to the overall distribution of condensed phase NR2B displacements (*Figure 3—figure supplement 3D3*). Switching probabilities from mobile to confined state ($P_{mc}$ = 0.59/frame) and from confined to mobile state ($P_{cm}$ = 0.09/frame) can also be readily determined from the number of H→L and L→H transition events based on the same confined/mobile (L/H) classification using $d_b$ = 60 nm. Notably, the fit to the overall condensed-phase displacement distribution in *Figure 3—figure supplement 3D3* is superior to that provided by the HMM-estimated two-state parameters (quality of the latter fit, with RMSE = 184.9, is comparable to that in *Figure 3—figure supplement 1C1* ). Because the HMM applied above seeks to optimize quantitative description of both displacements and switching probabilities while under the constraints imposed by the presumption of two simple diffusion states, its ability to accurately capture the distribution of displacements is likely limited despite its utility for estimating diffusion parameters.

## Generating simulated homogeneous and heterogeneous molecular systems

Molecules distributed homogeneously both in condensed and dilute phases were generated based on the Monte Carlo method to simulate localizations obtained in single-molecule tracking experiments. Diffusion coefficients $D$ and molecule densities $N$ were set for different scenarios. Averaged lifetime was set to 3.5 frames based on our experimental average track length and with a Poisson distribution. Number of total tracks was calculated before simulation and every track started with a random frame following uniform distribution. For one single track with diffusion coefficient $D$, the displacement in a short time interval $t$ (0.0001 s) will be

$$dx = dy = randn \times \sqrt{2Dt} \tag{4}$$

where $dx$ and $dy$ are the displacements along horizontal and vertical axes, and $randn$ is a random number with the standard normalized distribution. When a simulation step crosses two regions with different diffusion coefficients (e.g., dilute and condensed phases), the diffusion coefficient for the initial region is used for $D$ in *Equation 4* for the entire step (i.e., no change in the value of $D$ even if a single step crosses the boundary between two regions with different $D$ values). Because of the approximate empirical relation in *Equation 1*, this protocol is sufficient for ensuring that the simulations equilibrate to steady-state populations that resemble closely the equilibrium population distribution across different phases for the PSD system we considered (*Figure 3H*). All data were saved into two versions, one format included track information as ground truth and the other was formatted by frames and localizations in each frame without track information as simulated data for evaluate tracking algorithm.

Molecules distributed heterogeneously in condensed phases were generated with similar process with an additional set of switching probabilities between mobile and confined states. Molecules in a confined state will be restricted to a certain position, thus the simulated localizations will follow a Gaussian distribution based on the point spread function. A confined molecule can switch to mobile state in the next frame with a probability $P_{cm}$, and a mobile molecule can switch to a confined state in the next frame with a probability $P_{mc}$. Mobile ratio ($P_m$) can be calculated from the switching probabilities, viz., $P_m = P_{cm}/(P_{cm} + P_{mc})$. The lifetime of the mobile state can be expressed as $t_m = 1/P_{mc}$. The data were saved into two versions in a manner similar to that described for the homogeneous system.

Each scenario was simulated for 500–10,000 frames depending on the molecule density in order to obtain a similar total track number for every system. The script for the simulations was in-house coded by MATLAB.

## Analysis of simulated data

For each condition, simulated data were fed to the algorithm to assign localizations into tracks with different maximum step limit. According to *Equation 3*, the ratio of maximum step limit to root mean square displacement ($R/RMSD$, defined as $X$) is the key variable, so we covered $X$ from 1 to 5 with a step size of 0.5. Track assignment error composed of 'true negative tracks' (TN, two localizations belong to the same track in ground truth but not recognized as the same track) and 'false positive tracks' (FP, two localizations do not belong to the same track in ground truth but recognized as belonging to the same track) and calculated as the ratio of the absolute value of (the algorithm output – ground truth)/ground truth. Diffusion coefficient error was calculated by fitting MSD (for the homogeneous system) or fitting a two-state model (for the heterogenous system) and compared with the original setting for each condition. The scripts for the evaluations were coded by MATLAB.

## Simulation of Brownian motion model with motion-state switches

Consider a particle undergoing Brownian motion on a 2D plane with the additional feature that it can switch between confined and mobile states at each time step with certain switching probabilities. For the examples in *Figure 3E and F*, during each time step, if the particle is in a confined state, it will have a probability of $P_{cc} = 0.9$ to remain in the confined state and therefore a probability of $P_{cm} = 0.1$ to switch to the mobile state. If the particle is in the mobile state, it moves in a random direction with a random displacement (i.e., following the standard normal distribution) and, at the same time, possess a probability of $P_{mc} = P_{cm} P_c/P_m$ to switch back to the confined state. This set-up maintains a

steady-state confined-mobile balance because the total switch events from mobile to confined state and vice versa are identical, that is, $P_m P_{mc} = P_c P_{cm}$. Since $P_c + P_m = 1$, the probability of switching from mobile to confined state is $P_{mc} = P_{cm}(1-P_m)/P_m$. In *Figure 3F*, 500 particles and a simulation duration of T = 100 s are used in each simulation; the mobile ratio $P_m$ was studied from 0 to 0.9 with a step size of 0.1.

## Equilibrium simulation of coexisting phases and confined/mobile states

The phase boundaries were constructed in accordance with experiments (*Figure 1B*). The boundaries were not changed during the simulation. An area of 15 × 30 μm² with periodic boundary conditions was used for the simulation (*Figure 3—figure supplement 1D*). At the beginning of the simulation, a large number of molecules (50,000) were randomly distributed in the condensed and dilute phases with initial EF of 60. All molecules in dilute phase were treated as mobile during the simulation, for the test runs reported in *Table 1*, the diffusion coefficient was set at 0.6 μm²/s. The motions of molecules in condensed phase consisted of those in the confined state and the mobile state. Molecules in the confined state were treated as immobile with fixed positions. Molecules in the mobile state were undergoing Brownian motion and the diffusion coefficient was set at 0.1 μm²/s. The switch between confined state and mobile state was determined by the switching probability $P_{cm}$ and $P_{mc}$. $P_{mc}$ could be directly calculated through the averaged dwell time of mobile state (0.1 s) as the lifetime of the fluorophore (~1 s) was much longer than the molecule's dwell time. $P_{cm}$ was calculated using the equilibrium condition between the mobile and immobile state by

$$P_{mc} \times \eta = P_{cm} \times (1 - \eta)$$

where $\eta$ is the mobile ratio (10%). The simulation time step was t = 0.0001 s with the total simulation time being 100 s. The script for the simulation was in-house coded by MATLAB. In addition to the examples reported in *Table 1*, analogous simulations based upon diffusion coefficients extracted using HMM or our model-independent approach were also conducted as described in 'Results' of the main text.

## FRAP assay

Proteins were labeled with Alexa Flour 555 (Thermo Fisher) at 1% for PrLD-SAMME and PrLD-SAMWT or 0.5% for PrLD. FRAP assays were performed on a Zeiss LSM 880 confocal microscope at room temperature. A region for bleaching (R1) with a diameter of 2 μm was selected within a large, condensed droplet. A reference region (R2) with the same size of R1 was selected in another large, condensed droplet as the system control. R1 was bleached with 40/30/10 iterations with 100% 561 nm laser power and followed by recording fluorescence intensity of the selected regions for 100 s in the time-lapse mode with a 10 s gap between each point for the PrLD-SAMWT system and 5 s for PrLD-SAMME and PrLD systems. The fluorescence intensities were normalized to 0% right after photo-bleaching and to 100% before photo-bleaching. Each data point was calibrated by recorded fluctuation of the intensity of the reference region R2.

## FRAP simulation

FRAP simulations were based on the simulation for the equilibrium-state phase separation described above but with simplified model phase boundaries. A 20 × 8 μm² box with periodic boundary containing seven spherical condensed droplets was used (*Figure 5A*). The radius and center coordinates of condensed droplets were 0.5/1/1/2/1/1/0.5 μm and 1/3/5/10/15/17/19 μm from the left edge of the box, respectively. Conditions 1/2/3 (see *Figure 5A*) had the same bleaching size with a diameter of 1 μm and centered in the large/median/small droplets. The EF for the illustrative runs (*Figure 5B—E*) was set at 100 for all simulations, the mobile ratio was set at 100% or 10%, and the confined state lifetime was set at 10 or 100 s. For totally immobilized molecules, the lifetime of the confined state was infinite and the switching probability between mobile and immobile was zero. All simulations were carried out for 100 s with a time step of 0.0001 s. A total of 50,000 molecules were used for each simulation. In *Figure 5G*, instead of the above simulation parameters, diffusion coefficients and mobile ratios deduced from experimental measurements (*Figure 4*) were used for the simulation. The mobile state diffusion coefficient was 0.18/0.19/0.36 μm²/s for PrLD-SAMWT/PrLD-SAMME/PrLD simulation, respectively. The diffusion coefficient in dilute phase was 100 μm²/s. The

mobile ratios in the condensed phase were 26%/98.1%/99.6% for PrLD-SAMWT/PrLD-SAMME/PrLD, respectively. The region of the condensed phase was indicated as a circle with radius of 2/8/8 µm, respectively, for each simulation. The simulated FRAP region was at the center of the condensed phase region with a radius of 1 µm.

*Shen, 2023*

## Acknowledgements

This work was supported by grants from the National Natural Science Foundation of China (82188101), the Minister of Science and Technology of China (2019YFA0508402), Shenzhen Bay Laboratory (S201101002), RGC of Hong Kong (AoE-M09-12, 16104518 and 16101419), and a HFSP Research Grant (RGP0020/2019) to MZ. The research effort in HSC's group was supported by Canadian Institutes of Health Research grant PJT-155930 and Natural Sciences and Engineering Research Council of Canada grant RGPIN-2018-04351.

## Additional information

### Funding

| Funder | Grant reference number | Author |
| --- | --- | --- |
| National Natural Science Foundation of China | (82188101 | Mingjie Zhang |
| Ministry of Science and Technology | 2019YFA0508402 | Mingjie Zhang |
| Shenzhen Bay Laboratory | S201101002 | Mingjie Zhang |
| University Grants Committee | AoE-M09-12 | Mingjie Zhang |
| Human Frontier Science Program | RGP0020/2019 | Mingjie Zhang |
| Canadian Institutes of Health Research | PJT-155930 | Hue Sun Chan |
| Natural Sciences and Engineering Research Council of Canada | RGPIN-2018-04351 | Hue Sun Chan |
| University Grants Committee | 16101419 | Mingjie Zhang |
| University Grants Committee | 16104518 | Mingjie Zhang |

The funders had no role in study design, data collection and interpretation, or the decision to submit the work for publication.

### Author contributions

Zeyu Shen, Software, Formal analysis, Validation, Investigation, Methodology, Writing - original draft, Writing - review and editing; Bowen Jia, Investigation, Methodology; Yang Xu, Investigation; Jonas Wessén, Tanmoy Pal, Formal analysis, Validation; Hue Sun Chan, Formal analysis, Validation, Methodology, Writing - review and editing; Shengwang Du, Resources, Supervision, Methodology; Mingjie Zhang, Conceptualization, Resources, Formal analysis, Supervision, Funding acquisition, Methodology, Writing - original draft, Project administration, Writing - review and editing

### Author ORCIDs

Zeyu Shen http://orcid.org/0000-0002-1057-7191
Jonas Wessén http://orcid.org/0000-0002-5904-8442
Hue Sun Chan http://orcid.org/0000-0002-1381-923X
Mingjie Zhang http://orcid.org/0000-0001-9404-0190

Decision letter and Author response
Decision letter https://doi.org/10.7554/eLife.81907.sa1
Author response https://doi.org/10.7554/eLife.81907.sa2

## Additional files

### Supplementary files
• MDAR checklist

### Data availability

The home-written codes and use of the codes for data analysis described in this manuscript have been uploaded in the GitHub database with unrestricted access:https://github.com/NeoLShen/Code-for-phase-simulation-and-HMM-analysis, copy archieved at *Shen, 2023*.

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
