## [Editor Report]

In this work, the authors introduce an adaptive single-molecule tracking approach for following molecules within biomolecular condensates. Consistent with the emerging idea that condensates are not simple, purely viscous, Newtonian fluids, the authors find that the motions of molecules reveal intrinsic inhomogeneities switching between trapped and more mobile states that suggest the existence of at least two – possibly more – states within condensates. The data appear to be consistent with the formation of percolated networks within condensates – a finding that is likely to be general to other systems.

---

## [Decision Letter]

**Decision letter after peer review:**

Thank you for submitting your article "Biological condensates form percolated networks with molecular motion properties distinctly different from dilute solutions" for consideration by *eLife*. Your article has been reviewed by 2 peer reviewers, and the evaluation has been overseen by a Reviewing Editor and Aleksandra Walczak as the Senior Editor. The reviewers have opted to remain anonymous.

Essential revisions:

Both reviewers agree that the observations provide potentially important new information, but three specific concerns have been raised.

1) Is the parsing into weak, non-specific interactions driven by IDRs and strong, specific interactions driven by folded domains real or a trope? The authors propose that this is their belief. However, the specificity of interactions within IDRs, including the FUS system has been well established via the stickers and spacers framework. Conformational heterogeneity engenders conformational and hence concentration fluctuations. These effects are neither weak nor non-specific. This view is shared by the editor and reviewer 1. Therefore, instead of the artificial partitioning and the unsubstantiated claim that the network is small, it is better to update this perspective based on numerous recent contributions. Please see https://doi.org/10.1038/s41557-021-00840-w, and https://www.biorxiv.org/content/10.1101/2022.05.21.492916v2 as examples. The key point is that the canonical expectation is of shear thinning behavior in viscoelastic materials that have terminal viscous behaviors. Therefore, the issue is not about weak non-specific vs. strong, specific interactions, but more about the modes of motions accessible to folded domains vs. IDRs. Motions of the latter have been discussed extensively by the Schuler, Best, and Blackledge groups, and these motions span a spectrum of timescales and length scales, which explains the observations reported here, as opposed to the binary classification offered up by the authors and in the biochemical literature.

2) Reviewer 1 raises important issues regarding the simplicity of the 2-state model and the apparent cyclic nature of the parameterization, fitting, and analyses. Please respond constructively and completely to all the points raised by reviewer 1. A specific question that comes to mind from the comments of reviewer 1 is why won't a single stretched exponential with at least one less parameter than currently used provide an equivalent description of the data? One can imagine this will be true. And if so, how does something like a Kohlrausch-Williams-Watts function compare to the current approach?

3) Reviewer 2) raises several important points about the imposition of diffusive motions on the analysis. What are the criteria used to adjudicate in favor of diffusion and hence two diffusive processes? Further, the review notes that there are numerous accounts of caging that are described in the physical literature that is highly relevant in the current context.

*Reviewer #1 (Recommendations for the authors):*

1. Overall, the manuscript should review the relevant literature in the Introduction (at least to some extent) and evaluate the novelty of its own method based on this review. Some specific points are discussed below:

(p. 3) We need references for the sentence "The existing biochemistry and biophysics theories that have been guiding our understandings of molecular behaviours and their interactions in living cells in the past are mainly developed for molecules in dilute solutions." Indeed, there are very few references in the Introduction, which I don't think is appropriate.

(p. 3-4) There are several works trying to quantify the properties of condensates. For example, see 10.1016/j.bpj.2019.08.030 and 10.1038/s41467-020-19476-4; see also a recent review 10.1021/acs.chemrev.1c00774.

(p. 3-4) Also, there have been attempts to use super-resolution microscopy to track individual proteins (especially in vivo). I think that they should be introduced and appreciated. For example, see 10.1126/science.aar4199 and 10.1016/j.molcel.2020.06.034.

2. (p. 6) The authors assume that the phase boundaries are unchanging throughout their measurement. Is it justifiable?

3. (p. 6) They claim to estimate the diffusion coefficients of NR2B. Can they provide the values?

4. The two-state model, assuming trapped and mobile states of molecular diffusion, is a simple and powerful model, but I don't think it necessarily reflects real physics. In my opinion, it is equally likely that the molecules have three or four possible states, or even a continuum of mobility, as the distributions (Figure 3) do not show any "distinct" peaks. Can the authors comment on this?

5. Figure 3A1 and 3B: the authors conclude that a simple diffusion model is a "bad" fit for Figure 3A1 while it explains Figure 3B well – but are they really different? R2 is even better for Figure 3A1 (0.97 vs. 0.91).

6. In their model, they use the "switching probabilities," which in my opinion is not necessary. In a dynamic equilibrium (as in the plateaus of Figure 3C and 3D), the system should be in a quasi-static state, where we can simply assume two diffusion behaviors even without considering their exchange. Can we simply use a linear combination of two simple diffusion distributions to fit the data?

7. I am also curious about the performance of the HMM. How much does it improve the fitting? Can the authors provide the fitting results?

8. (p. 8) The authors claim that the diffusion coefficient of NR2B in a mobile state in the condensed phase and that of NR2B in the dilute phase are "very close," but the reported values are ~0.47 μm2/s and ~0.61 μm2/s. The authors should clarify in what sense they are very close to each other.

9. (p. 8) The authors mention that they did not observe any obvious hinderance against motions when molecules cross the phase boundaries (and hence assume that the flux is simply the diffusion coefficient multiplied by molecular density), but there are reports on the diffusion barrier at the boundaries (see 10.1038/nature22989 and 10.3390/ncrna5040050). In my opinion, the barrier makes sense, as the surface tension will break the symmetry between one side and the other side of the boundary. Can the authors comment on this?

10. (p. 9) The theoretical value for the enrichment fold is 62.8 and its experimental value is ~61, which may look astonishing, but the "theoretical" value is not calculated from scratch. It is based on the parameters that fit the data, so I think it is unsurprising that they obtain very similar values. I would be surprised if there was a big discrepancy.

11. (p. 9) Again, they report the "remarkable" similarity between the diffusion coefficient for the mobile fraction in the condensed phase and that in the dilute phase, but I think the difference (0.17 vs 0.47 μm2/s) can be considered significant, depending on your perspective.

12. The authors conducted a Monte Carlo simulation to obtain the simulated enrichment fold. Is it just for validation of the analytical formula? (If this is the case, I don't think it is necessary to include the simulation results.) Can we obtain any other useful information from the simulation?

13. In Figure 4A and the corresponding text, the authors seem to claim that "weak" interactions will lead to "small" networks, but I think that this statement can be misleading. It may be true that molecules are involved in small networks at a certain time point, but the interactions are transient and dynamic (as the authors mention), so molecules change their partners rapidly and on average, the molecules are involved in a large (if not system-spanning) network. Hence, the "size" of the network should be discussed with care, and I recommend the authors revise the manuscript accordingly.

14. (p. 11) The authors say that the fraction of dwell time is "directly proportional" to the binding affinity and avidity, which I don't think is a mathematically precise statement. Did they mean "directly dependent on the two factors"?

15. If I understand correctly, the FUS experiment was conducted in 3D, unlike the NR2B experiment, where the molecules are attached to the membrane. As the system becomes three-dimensional, the "displacement" is now a 2D-projected value, and we need to devise a way to convert it to a 3D value. Do the authors consider this point? If so, please provide a description of their conversion method.

16. (p. 12) The authors say that phase separation of FUS PrLD took up to 12 hours to occur but do not show any data. Please include the data (microscope images or turbidity data).

17. (p. 15) Can the authors comment on why they have greater error bars for simulated FRAP curves in Figure 5G? Is it because they have a smaller number of measurements (3 vs. 10)?

*Reviewer #2 (Recommendations for the authors):*

The central finding that the molecules tend to experience transiently confined states in the condensed phase is remarkable and important. This finding is reminiscent of transient "caging"/"trapping" dynamics observed in diverse other crowded and confined systems e.g., https://doi.org/10.1103%2FPhysRevLett.107.178103, https://doi.org/10.1038/s41467-019-10115-1, https://doi.org/10.1103%2FPhysRevLett.89.095704, https://doi.org/10.1103%2FPhysRevLett.92.178101, https://doi.org/10.1529%2Fbiophysj.106.092619, https://doi.org/10.1016%2Fj.bpj.2013.12.013. The authors may wish to comment on these conceptual connections to other systems that highlight the broader context of this fascinating finding; it might motivate others to bring theoretical and analytical approaches developed to understand these other systems to bear on condensates, which would be valuable to the field.

Related to the previous point: it would be interesting to see not just the distribution of displacements, but also the distribution of times spent in the confined state and mobile state. Given the experimental results, the authors likely already have these data. The functional form of this distribution is known to reflect the physics underlying the trapping behavior and transitions between the two states (see e.g., https://doi.org/10.1016/0370-1573(90)90099-N).

Also related to the previous point: it is very surprising to see the authors interpret the single-molecule motion as being 'normal' diffusion (within the context of a two-state diffusion model), instead of analyzing their data within the context of continuous time random walks or anomalous diffusion, which is generally known to arise from transient trapping in crowded/confined systems (again see e.g., https://doi.org/10.1103%2FPhysRevLett.107.178103, https://doi.org/10.1038/s41467-019-10115-1, https://doi.org/10.1103%2FPhysRevLett.89.095704, https://doi.org/10.1103%2FPhysRevLett.92.178101, https://doi.org/10.1529%2Fbiophysj.106.092619, https://doi.org/10.1016%2Fj.bpj.2013.12.013). It is not clear that interpreting the results within the context of simple diffusion is appropriate, given their general finding of the two confined and mobile states. Such a process of transient trapping/confinement is known to lead to transient subdiffusion at short times and then diffusive behavior at sufficiently long times. There is a hint of this in the inset to Figure 3, but these data need to be shown on log-log axes to be clearly interpreted. I encourage the authors to think more carefully and critically about the nature of the diffusive model to be used to interpret their results.

---

## [Author Response]

Essential revisions:Both reviewers agree that the observations provide potentially important new information, but three specific concerns have been raised.1) Is the parsing into weak, non-specific interactions driven by IDRs and strong, specific interactions driven by folded domains real or a trope? The authors propose that this is their belief. However, the specificity of interactions within IDRs, including the FUS system has been well established via the stickers and spacers framework. Conformational heterogeneity engenders conformational and hence concentration fluctuations. These effects are neither weak nor non-specific. This view is shared by the editor and reviewer 1. Therefore, instead of the artificial partitioning and the unsubstantiated claim that the network is small, it is better to update this perspective based on numerous recent contributions. Please see https://doi.org/10.1038/s41557-021-00840-w, and https://www.biorxiv.org/content/10.1101/2022.05.21.492916v2 as examples. The key point is that the canonical expectation is of shear thinning behavior in viscoelastic materials that have terminal viscous behaviors. Therefore, the issue is not about weak non-specific vs. strong, specific interactions, but more about the modes of motions accessible to folded domains vs. IDRs. Motions of the latter have been discussed extensively by the Schuler, Best, and Blackledge groups, and these motions span a spectrum of timescales and length scales, which explains the observations reported here, as opposed to the binary classification offered up by the authors and in the biochemical literature.

We thank the comments from the editor and the reviewer 1. Their points are well taken, and the two references mentioned above are now included to broaden the perspectives discussed in the revised manuscript.

We understand that our hypothesis and description of supporting evidence for binding affinity and specificity of IDRs may be contentious and will likely draw criticisms from many colleagues in the field. We also understand that binding affinity and degree of specificity of biomolecules are relative terms and therefore can be misunderstood if taken out of context (i.e., away from biological settings). Nonetheless, binding affinity of a molecular interaction system can be quantified by measuring the binding constant of the system. There are strong and there are weak interactions and their physical effects are different. On the other hand, quantification of binding specificity, though possible, is much trickier. Binding specificity of an interaction is affected by factors in addition to the binding affinity and protein concentrations in a given reaction system.

To better address this issue, we have performed additional experiments and reported the new results in Figure 4 —figure supplement 1 (under the heading “Dynamic molecular networks in condensed phases” in the revised manuscript).

In the system described in this additional work, we are comparing a classical IDR protein fragment, the PrLD of FUS, with the molecular interactions between folded proteins/domain and their targets. We measured the binding affinity of FUS PrLD using two different approaches. First, we used analytical size exclusion chromatography coupled with static light scattering (SEC-SLS) to assay the interaction between FUS PrLD. On SEC column, GB1-tagged PrLD (theoretical M.W. 37.0 kDa) was eluted at a molecular mass corresponding to a monomer when the loading concentrations of the proteins were at as high as 100 μM (blue curve, measured MW 34.3±0.5 kDa) or 300 μM (green curve, measured MW 35.0±0.9 kDa) (please see Figure 4–figure supplement 1A). To rule out potential impact of the GB1 tag on the FUS PrLD interaction, we cleaved the GB1 tag. We took the advantage that tag cleaved FUS PrLD was stable and monodispersed in solution for up to ~10 hrs before phase separation occurs (see Figure 4–figure supplement 2), so we performed SEC-SLS assay of FUS PrLD with the GB1 tag freshly cleaved. The theoretical MW of tag-free FUS PrLD is 22.3 kDa. The measured MW for the tag-free FUS PrLD at the loading concentration of 100 μM (blue curve, measured MW 21.8±3.5 kDa) or 300 μM (green curve. measured MW 20.9±1.1 kDa) also corresponds to the monomer state of the protein (Figure 4–figure supplement 1B). As a control, the PrLD-SAMWT fusion protein (theoretical MW 80.7 kDa) was eluted as an oligomer and its elution volumes were heavily dependent on the loading concentrations (loading concentration at 30 μM, blue curve, measured MW 165.0±0.4 kDa); loading concentration at 150 μM, green curve; measured (MW 265.0±0.4 kDa) (Figure 4–figure supplement 1C). The SEC-SLS analysis results shown in the Figure 4–figure supplement 1A-C demonstrate that FUS PrLD displays a VERY weak inter-molecular interaction (i.e., near the detection limit of the assay method). One may argue that the nearly no detectable interaction between FUS PrLD on analytical size exclusion column could arise from unique kinetic behaviours of the disordered protein (e.g., fast off-rates due to high conformational dynamics of the IDR). We next resorted to a thermodynamic-based binding assay using isothermal titration calorimetry (ITC). We titrated 300 μM FUS PrLD (either without or with the GB1 tag cleaved) into 30 μM FUS PrLD in the reaction cell. Again, the ITC-based assay showed that nearly no detectable interactions could be observed between FUS PrLD (Figure 4—figure supplement 1).

The results presented in the Figure 4—figure supplement 1 demonstrate that the interaction between FUS PrLD is very weak, with a K_d_ value larger than a few hundreds μM based on the experimental methods used here. All of the above information about the new experimental measurement is now included as a “Data interpretation” paragraph after Figure 4 —figure supplement 1 of the revised manuscript. It is thermodynamically challenging to achieve highly specific binding of FUS PrLD to itself than to other proteins existing in the proteome of a living system if the self-association binding constant is extremely low (i.e., the percentage of productive FUS PrLD complex formation at the cellular protein concentration would be very low). Because of the experimental observations (including the above described additional data) and our narrative arguments, intellectual honesty demands us to retain our statement about the relative weak interactions between IDRs. Clearly, with the materials provided by our manuscript, including its supplementary information and review documents, our perspective on this point may attract future research on this important topic. In our estimation, such evidence-based debates and open airing of opinions on a yet-to-be-resolved issue are healthy and can help science to progress. Indeed, in the full-length FUS, its IDRs likely function together with its folded domains (the specific RNA binding RRM domain and the zinc finger) that are possibly involved in stronger and more specific interactions to achieve its cellular functions.

We certainly recognize that biological systems are complex and have always been giving us surprises that fall outside our common wisdom. Therefore, we have removed the statement “it is our opinion that most cellular condensates belong to such category” from the description “For biological condensates of which their formations are largely driven by specific molecular interactions (it is our opinion that most cellular condensates belong to such category), molecules in the condensed phase spend most of their time in the confined state”. In line with the editor’s comments, we further calibrated our description by removing the term “non-specific” in our description of IDR-based interactions, as we indeed do not have experimental data to characterize the specificity of the interaction between FUS PrLD (and also because of trickiness in defining binding specificity of a biomolecular interaction). Finally, in case a misunderstanding exists, we wish to emphasize that it is not our intention to under appreciate roles of IDRs in phase separation. Instead, we wish to point out that combinations of strong and specific interactions with generally weak and IDR based interactions can lead to formation of biological condensates with very broad and distinct molecular properties. To help put our perspective in a more comprehensive context, we have now included the aforementioned two recent references from the laboratories of Mittag and Pappu in our introductory discussion.

2) Reviewer 1 raises important issues regarding the simplicity of the 2-state model and the apparent cyclic nature of the parameterization, fitting, and analyses. Please respond constructively and completely to all the points raised by reviewer 1. A specific question that comes to mind from the comments of reviewer 1 is why won't a single stretched exponential with at least one less parameter than currently used provide an equivalent description of the data? One can imagine this will be true. And if so, how does something like a Kohlrausch-Williams-Watts function compare to the current approach?

Please refer to our response to comments point 4 and 7 from reviewer 1.

3) Reviewer 2) raises several important points about the imposition of diffusive motions on the analysis. What are the criteria used to adjudicate in favor of diffusion and hence two diffusive processes? Further, the review notes that there are numerous accounts of caging that are described in the physical literature that is highly relevant in the current context.

Please refer to our detailed responses to comment point 1 from reviewer 2.

Reviewer #1 (Recommendations for the authors):1. Overall, the manuscript should review the relevant literature in the Introduction (at least to some extent) and evaluate the novelty of its own method based on this review. Some specific points are discussed below:(p. 3) We need references for the sentence "The existing biochemistry and biophysics theories that have been guiding our understandings of molecular behaviours and their interactions in living cells in the past are mainly developed for molecules in dilute solutions." Indeed, there are very few references in the Introduction, which I don't think is appropriate.(p. 3-4) There are several works trying to quantify the properties of condensates. For example, see 10.1016/j.bpj.2019.08.030 and 10.1038/s41467-020-19476-4; see also a recent review 10.1021/acs.chemrev.1c00774.(p. 3-4) Also, there have been attempts to use super-resolution microscopy to track individual proteins (especially in vivo). I think that they should be introduced and appreciated. For example, see 10.1126/science.aar4199 and 10.1016/j.molcel.2020.06.034.

We thank the reviewer for pointing out the deficiency in previous introductory discussion and his/her constructive suggestions. We have now included a series of references on the quantitative characterizations of condensates in the revised manuscript with brief descriptions. We have also included a description of super-resolution microscopy-based tracking studies of proteins in living cells with proper citations. The added text reads as “A biological condensate formed via phase separation is more of a condensed soft matter system, thus theories dealing with dilute solutions are not expected to be generally adequate for condensed molecular systems. Several quantified approaches were developed to study the properties of condensate (Abyzov et al., 2022; Song et al., 2020; Taylor et al., 2019). Super-resolution and single molecule tracking based method was also used to characterize the biophysical properties of biological condensates (Cho et al., 2018; Garcia et al., 2021; Guilhas et al., 2020; Kent et al., 2020; Moon et al., 2019; Muñoz-Gil et al., 2022)”. Note that in addition to the (Abyzov et al., 2022; Cho et al., 2018; Guilhas et al., 2020; Song et al., 2020; Taylor et al., 2019) references suggested by the reviewer, we have also included new references (Garcia et al., 2021; Kent et al., 2020; Moon et al., 2019; Muñoz-Gil et al., 2022; Muñoz-Gil et al., 2021) in the paragraph in the revised manuscript quoted above.

2. (p. 6) The authors assume that the phase boundaries are unchanging throughout their measurement. Is it justifiable?

This is indeed a valid and highly relevant concern. The boundaries of the condensed phases in our experimental system do change, as processes like Oswald ripening and droplet fusions (e.g., very small droplet below our optical detection fusing with a nearby large droplet) must be happening throughout our experimental measurements. In our experiments, we chose a relatively short time period for our measurement (60 seconds). We compared the phase boundaries of the systems before and right after the super-resolution imaging session. The overall phase boundary did not undergo obvious change during the imaging process as the shapes and boundaries of the condensed droplets in the system remained essentially the same (see Figure 1—figure supplement 1). Thus, we anticipate that the error introduced by assuming the constant phase boundary during the super-resolution imaging process should not be large.

3. (p. 6) They claim to estimate the diffusion coefficients of NR2B. Can they provide the values?

The initial rough estimation of diffusion coefficient of NR2B was based on single molecule tracking results with default search range (500nm) in consecutive frames in both condensed and dilute phases. This procedure is for ascertaining optimal search ranges for subsequent analyses of experimental data. We then fit the displacement distributions in dilute phase with a 2D simple diffusion model and fit the tracks in condensed phase with a hidden Markov model (HMM) to estimate the diffusion coefficients in both phases first. With these rough estimates, we then determined the optimized search range for each phase. For the case in the manuscript, the estimated diffusion coefficients of NR2B in condensed phase and dilute phase are 0.22 μm^2^/s and 0.57 μm^2^/s, respectively. We have included this description in the main text part of the revised manuscript (p.7 of the revised manuscript) to make our procedure clearer: “… determining optimized search range for different phases in subsequent analyses. The initial roughly estimated overall diffusion coefficients of NR2B in condensed phase and dilute phase are 0.22 μm^2^/s and 0.57 μm^2^/s, respectively” as well as referred to the detailed description in Materials and methods.

4. The two-state model, assuming trapped and mobile states of molecular diffusion, is a simple and powerful model, but I don't think it necessarily reflects real physics. In my opinion, it is equally likely that the molecules have three or four possible states, or even a continuum of mobility, as the distributions (Figure 3) do not show any "distinct" peaks. Can the authors comment on this?

We agree with the reviewer’s general tenet on this point. In principle, molecules in the condensed phase could have multiple motion states instead of a simple two-state model described as a first approximation in our manuscript. For instance, one might envision that the percolated molecular network in the condensed phase is not uniform (e.g., existence of locally denser or looser local networks) and dynamic (i.e., local network breaking and forming). Therefore, individual proteins binding to different subregions of the network will have different motion properties/states. To emphasize this understanding and to ensure that our simple two-state model—which is intended (and now stated clearly) as a first approximation—will not be misconstrued as the be-all and end-all physical picture, we have now expanded the pertinent discussion with the addition of a schematic figure for the underlying energy landscape of a phase-separated system (Figure 3G in the revised manuscript). Physically, as the reviewer has correctly pointed out, the motion states of molecules in the percolated condensed phase are most likely a continuum of mobilities. Our described “confined state” in the manuscript represents an ensemble average of such continuum mobilities. We have now emphasized this point at multiple places in the revised manuscript. We thank the reviewer for this very incisive comment.

5. Figure 3A1 and 3B: the authors conclude that a simple diffusion model is a "bad" fit for Figure 3A1 while it explains Figure 3B well – but are they really different? R2 is even better for Figure 3A1 (0.97 vs. 0.91).

Thanks for the observation. Indeed, if one simply looks at the square Perason coefficient value (R2, denoted as *r*^2^ in the revised manuscript), fitting the molecular displacements of the condensed phase in Figure 3A1 by a simple diffusion model is apparently good; but comparison of the best fitted curve with the experimental distribution (Figure 3 —figure supplement 1C1) reveals significant mismatches for high displacements. Since the great majority of the condensed-phase molecules are in the confined state, the simple diffusion based model fitting is naturally skewed to these confined molecules, leaving fitting (or rather poor fitting) of motions of the small proportion of mobile molecules insignificant to the R2 value of the fitting (i.e., the motions of mobile state molecules are neglected in the fitting, see Figure 3—figure supplement 1C). In this regard, the match between the fitted curve and experimental displacement distribution in Figure 3B is significantly superior by comparison. We have now underscored this observation in the sentence “Fitting the histogram with a single population of NR2B undergoing Brownian motion could only cover the confined state peak of molecule but not the high-displacement tail of this distribution (Figure 3—figure supplement 1C1). By comparison, in the dilute phase, the overall displacements of NR2B are much larger as well as broader and there is no prominent peak at very small displacements (Figure 3A2)” (p.8 of revised manuscript). In other words, R2 per se is insufficient for fully characterizing the quality of the fits in this situation. Indeed, the properties of condensed-phase mobile molecules are key for understanding the molecular behaviours of the condensed phase, and thus motions of this portion of molecules will need to be tackled separately. We have now emphasized this point in our revised manuscript (p.8 of the revised manuscript). Please see also our new model-independent analysis that does not presume simple diffusion processes as detailed below in our response to this reviewer’s point #7.

6. In their model, they use the "switching probabilities," which in my opinion is not necessary. In a dynamic equilibrium (as in the plateaus of Figure 3C and 3D), the system should be in a quasi-static state, where we can simply assume two diffusion behaviors even without considering their exchange. Can we simply use a linear combination of two simple diffusion distributions to fit the data?

This is a valid comment. We agree with the reviewer that if we only consider the equilibrium state (at effectively infinite time), the displacement distribution will be independent of the switching probabilities. However, the switching probabilities do affect the overall displacement distribution at finite time. Regardless, for our present interest, switching probabilities are essential parameters characterizing the kinetic rate of a molecule converting between the confined and mobile states, as there is no good physical reason to believe that a molecule in the confined state or mobile state will, respectively, remain forever confined or mobile. Specifically, these parameters are critically important when we investigate kinetic properties in non-equilibrium processes such as FRAP experiments. As we have shown in Figure 5, molecules with the same enrichment fold and the same mobile/confined state ratio but with different confined state lifetime (i.e., different “switching probabilities”) will have very different recovery kinetics. In our perspective, confined-to-mobile switch probability is correlated with the off-rate of a molecule dissociating from the percolated network. Thus, including switching probabilities is necessary for obtaining information of interest from analysing the single molecule tracking data in our study.

7. I am also curious about the performance of the HMM. How much does it improve the fitting? Can the authors provide the fitting results?

The overall displacement distribution improved only slightly using HMM results (R2=0.98 vs R2=0.97 in simple 2D Brownian motion distribution with a small decrease in RMSE, see Figure 3—figure supplement 1C1 and Author response image 1). One possible reason behind the apparent inability of HMM to provide a significant better fit of the displacement data is that the HMM model aims not only to optimize fitting with the displacements but also with the relationship between consecutive steps in terms of switching probabilities. In other words, the HMM applied here seeks to optimize quantitative description of both displacements and switching probabilities under the constraints imposed by the presumption of two simple diffusion states. It is likely that these constraints limit the ability of the present HMM to capture the distribution of displacements with high accuracy. Nonetheless, the diffusion rates estimated by HMM are not dramatically different from those estimated from a model-independent approach that achieves a significant better fit for the displacement distribution in the condensed phase (see Author response image 1). With these observations in mind, we regard the present HMM construct as an efficient method for yielding approximate diffusion coefficients and employ it in conjunction with our newly developed model-independent approach to offer a semi-quantitative yet useful description of our experimental data. To highlight this rationale and in recognition of the limitation of HMM, we now characterize our analysis as “semi-quantitative” (p.14 of the revised manuscript), and stated explicitly on p.32 of the revised manuscript that “the fit to the overall condensed-phase displacement distribution in Figure 3 —figure supplement 3D3 is superior to that provided by the HMM-estimated two-state parameters (quality of the latter fit, with RMSE = 184.9, is comparable to that in Figure 3 —figure supplement 1C1). Because the HMM applied above seeks to optimize quantitative description of both displacements and switching probabilities while under the constraints imposed by the presumption of two simple diffusion states, its ability to accurately capture the distribution of displacements is likely limited despite its utility for estimating diffusion parameters.”

**Author response image 1. sa2fig1:** Comparing fitting the experimental molecular displacements with a simple Brownian diffusion model (Figure 3—figure supplement 1C1) against fitting with the optimized HMM two-state diffusion model. Best fit of the same experimental displacement distribution in condensed phase with a two-state HMM model. Red Curve is the HMM model fit obtained by non-linear least squares method using MATLAB. R^2^ = 0.98, RMSE = 184.9..

Because of the limitations of our HMM analysis as noted above, and more fundamentally because of our general concern (as also voiced by the editor and the two reviewers) that presumption of a particular type of diffusive motion—in our case two-state simple Brownian diffusion—might engender biases, during this revision we endeavoured to develop a relative model-independent analysis of the time scales of our experimental displacements. The new results are reported in Figure 3 —figure supplement 3 and corresponding text in the “Results” (p.14-p.15) and “Materials and methods” (p.30p.32) sections of the revised manuscript. In addition to providing a sound basis for the existence of (at least) two significant different time scales in our experimentally observed tracks in the PSD condensed phase (see Figure 3—figure supplement 3), the new treatment also affords a fit of the experimental condensed-phase displacement distribution in terms of two diffusion processes (Figure 3—figure supplement 3D3) that is significantly better than that provided by HMM (Author response image 1, as already noted). Because both our HMM and model independent analysis consistently point to the existence of two significantly different time scales in condensed-phase NR2B motion with estimated diffusion coefficients and other motion parameters sharing a similar trend though not identical (see specific information below), we deem our hypothesis that NR2B motion in PSD condensed phase is physically divided into a broadly construed confined state and a broadly construed mobile state is well supported. Nonetheless, we do recognize that there are ample rooms for further development to improve our quantitative analysis (as noted below), an effort that is beyond the scope of this manuscript but should be taken up in future investigations.

To highlight this perspective and the prospects of future work, the following modified paragraph with three additional references on theoretical/computational analyses of complex diffusion processes is now included in the “Discussion” section of the revised manuscript:

“We have demonstrated that a two-motion-state model can provide a semi-quantitative rationalization of our experimental data and, at the same time, offer a conceptual framework for their physical interpretation. Indeed, such a two-motion-state picture is well motivated by direct experimental observations (Figure 2E, Video 1). Nonetheless, given the inherent complexity of biological condensates, it is intuitive to recognize that molecules in the condensed phase corresponding to a biological condensate would most likely have multiple motion states due to the heterogeneity of intra-condensate interactions (cf. schematics in Figure 3G). With this in mind, the two motion states in our simple two-state model for condensed-phase dynamics should be understood to be consisting of multiple sub-states. For instance, one might envision that the percolated molecular network in the condensed phase is not uniform (e.g., existence of locally denser or looser local networks) and dynamic (i.e., local network breaking and forming). Therefore, individual proteins binding to different sub-regions of the network will have different motion properties/states. It follows that molecular motions in the percolated condensed phase likely cover a continuum of mobilities, although the confined- and mobile-state diffusion coefficients deduced from our experimental measurements based on the two state model do represent two important time scales of intra-condensate dynamics of PSD. In light of this basic understanding, the “confined state” and “mobile state” as well as the derived diffusion coefficients in this work should be understood as reflections of ensemble averaged properties arising from such an underlying continuum of mobilities. Further development of experimental techniques in conjunction with more refined models of anomalous diffusion (Joo et al., 2020; Kuhn et al., 2021; Muñoz-Gil et al., 2021) will be necessary to characterize these more subtle dynamic properties and to ascertain their physical origins.” (p.22-p.23 of the revised manuscript).

The following subsection describing the main results and implications of the new model independent approach is now added to the “Results” section of the revised manuscript (with two added references on Lévy distribution):

“Classification of molecular displacements without presuming simple diffusion models

It should be emphasized that while quantitative aspects of this two-state perspective of condensed-phase NR2B dynamics was initially deduced by using a HMM assuming two simple diffusion processes, the observation of a general separation of time scales into roughly a low-mobility (L) confined state and high-mobility (H) mobile state in the model PSD condensate is intuitive (Movie 1-supplemental movie) and can be quantified without presuming simple diffusion. This is showcased by a “model-independent” formulation we developed here to classify experimental displacement steps based on a criterion on the correlation of the magnitudes of consecutive displacement steps in each experimental track. As detailed in Materials and methods, we found that the magnitudes of displacements d*_i_* of consecutive steps (experimental time frames) in condensed-phase NR2B are correlated, indicating clearly the process is not simple Brownian diffusion. Analysis of conditional displacement distributions (subsets of overall distribution) based on putative demarcation values (d_b_) for d_i_ suggests that d_b_ = 60 nm provides a reasonable classification of displacement steps into two broad classes, namely a low-mobility (L) state and a high mobility (H) state that correspond roughly to the above HMM-deduced confined and mobile states (Figure 3—figure supplement 3). The dynamic complexity of the PSD condensed phase is underscored by the observation in Figure 3—figure supplement 3D1,D2 that the displacement distributions of both the L and H states exhibit deviations from simple Brownian diffusion, with the deviation more appreciable for the H states, which are better fitted by a Lévy distribution (Viswanathan et al., 2008; Wang et al., 2020). Nonetheless, as a first approximation, L-state and H-state motion can be reasonably fitted by two separate simple diffusion process to provide a semi-quantitative description of our experimental data. In this simplified framework, the confined (L) state distribution was interpreted as caused by detection error whereas the mobile (H) state was described approximately by a simple Brownian diffusion process with diffusion coefficient *D*_m_=0.044 ± 0.05 μm^2^/s (Figure 3—figure supplement 3D). As specified in Materials and methods, the corresponding confined and mobile ratios and switching probabilities between the two motion states were readily obtained, with the mobile ratio P_m_ = 13.5%. Together with the dilute-phase *D*_d_ = 0.47 μm^2^/s estimated above, the enrichment fold estimated using eq.1 is EF ≈ *D*_d_/ P_m_*D*_m_ ≈ 79.

Applying these parameters (instead of HMM parameters) to the above-described Monte Carlo simulation using experimental phase boundaries yielded an EF value of 76.2±0.6. In view of the aforementioned approximate nature of the fitted simple diffusion process for mobile-state motion and that the *D*_m_ and P_m_ values estimated from Figure 3—figure supplement 3 are appreciably different from those estimated by HMM, the EF value of ≈76 from the present model-free, correlation-based consideration is seen as reasonably close to the experimental value of EF ≈ 61, thus attesting to the robustness of the physical picture that NR2B diffusion in the PSD condensates is roughly separable into two dynamic time scales. However, as stated above, the confined state and the mobile state each likely represents a combination of many sub-states due to complex molecular network formation in the condensed phase. The nature of these sub-states deserves further experimental investigations and more refined theoretical modeling in future studies.

In any event, taking all the above experimental data and computational analysis together, we now have a workable physical picture for molecular diffusion in the equilibrium state of a phase separation system. Our conceptual framework explicitly connects a set of measurable microscopic molecular motion properties with the observable macroscopic parameters of molecules in the system. The experimental method and the theoretical models developed above are robust and simple to implement. They should be useful for analyzing biomolecular phase separations in general.” (p.14-p.15 of the revised manuscript; verbatim quote except “Figure 3 —figure supplement 3” in the revised manuscript).

The following subsection providing technical details of the new model-independent approach is now added to the “Materials and methods” section of the revised manuscript:

“Correlation-based classification of high- and low-mobility displacement steps without presuming simple diffusion.

Because presumption of a motion type such as a combination of simple Brownian diffusions may artificially introduce an unwarranted separation of time scales and other possible biases, we developed a model-independent approach to analyse our experimentally determined displacements (Figure 3—figure supplement 3). To gain quantitative physical insights into NR2B molecular motion within the condensed phase, we consider a total of 2,522 experimental trajectories that are configured entirely in the condensed phase (i.e., no transition to the dilute phase) and each consists of at least 10 steps (i.e. 11 positions). The total number of displacements d*_i_* in this set of trajectories is 41,854, with an overall distribution P(d*_i_*) shown by the black curve in Figure 3—figure supplement 3A. To probe the extent to which NR2B motion in PSD condensed phase deviates from simple diffusion, we divided the experimental displacement into two classes, namely d*_i_* < d_b_ and d*_i_* > d_b_, using a boundary or demarcation d_b_ between small and large displacements (an optimized value of which is to be determined; note that there is no d*_i_* = d_b_ for the d_b_ values we considered) and then obtained two conditional distributions of displacements P(d_*i*+1_|d_i_ < d_b_) and P(d_*i*+1_|d*_i_* > d_b_) for d_*i*+1_ given that the preceding displacement d*_i_* is, respectively, less than or larger than d_b_ (see, e.g., the blue and orange histograms in Figure 3—figure supplement 3A for d_b_ = 60 nm). If overall condensed-phase NR2B motion were a simple diffusion process, the distribution of displacement of a given step should be independent of the displacement of the previous step and thus the two conditional distributions should be identical; but Figure 3—figure supplement 3A demonstrates that the two conditional distributions are significantly different, indicating once again that condensed-phase NR2B motion cannot be described as a simple diffusion process.

This observation from Figure 3—figure supplement 3A suggests that we may, as a first approximation, attempt to classify condensed-phase NR2B motion into a low-mobility (L) and a high mobility (H) states by using a demarcation d_b_ that results in maximal difference between the low- and high-mobility conditional displacement distributions. Here we use the overlap coefficient (denoted OVL) to quantify the L-H difference in conditional displacement distribution. In general, OVL is a measure of how much two distributions overlap, viz. OVL = ∫ d*x* min[*P*(*x*), *Q*(*x*)], where *P*(*x*) and *Q*(*x*) are the distributions of interest (i.e., OVL is the area of the overlapping region of the two distributions; see, e.g., (Song et al., 2017)). We computed OVL among the overall displacement distribution, the L and H conditional displacement distributions and found that the OVL between the H and L conditional displacement distributions reaches its minimum (i.e., with maximally difference) at around d_b_ = 60 nm (Figure 3—figure supplement 3B). We therefore adopt d_b_ = 60 nm as a workable demarcation between low- and high-mobility states of NR2B motion in the model PSD condensate. This classification is illustrated by the example provided in Figure 3—figure supplement 3C. In general, among a set of ~40 randomly selected experimental trajectories we inspected in detail, a classification using d_b_ = 60 nm appears reasonable although in some cases some of the relatively short displacements are slightly longer than 60 nm. We may then proceed to use d_b_ = 60 nm to resolve the overall displacement distribution into a low-mobility (L) component (Figure 3—figure supplement 3D1) and a high-mobility (H) component (Figure 3—figure supplement 3D2). As shown in the figure, the L-component distribution may be reasonably fitted (albeit with some deviations) to a simple-diffusion functional form with a parameter *s* = 13.6 ± 3.7 nm, where *s* may be interpreted simply as a detection error or alternately expressed as *s* = *D*_L_t with *D*_L_ = 0.006149 μm^2^/s being the fitted confined-state diffusion coefficient and t = 0.03s is the time interval of the time step between experimental frames.

If we fit the H-component distribution to a simple diffusion process, the best fitted diffusion coefficient for the mobile state is *D*_m_=0.044 ± 0.05 μm^2^/s (blue curve in Figure 3—figure supplement 3D2). Although a Lévy distribution (Viswanathan et al., 2008; Wang et al., 2020) for displacements in anomalous diffusion (Bouchaud and Georges, 1990; Joo et al., 2020) offers a clearly superior fit (orange curve in Figure 3—figure supplement 3D2) and should be explored for more detailed analysis of the H- as well as L-component displacement distributions in future studies, it suffice for our present purpose to limit our discussion of condensed-phase motion largely to a combination of simple diffusion processes depicted by the fitted blue curves in Figure 3—figure supplement 3D1,D2. In that case, with the confinement ratio (P_c_ = 86.5%) and mobile ratio (P_m_ = 13.5%) provided directly by the number of displacements classified as “L” and “H” in accordance with a d_b_ = 60 nm demarcation, the combination of the two fitted simple diffusion processes is seen to provide a reasonably good approximation to the overall distribution of condensed phase NR2B displacements (Figure 3—figure supplement 3D3). Switching probabilities from mobile to confined state (P_mc_ = 0.59 / frame) and from confined to mobile state (P_cm_ = 0.09 / frame) can also be readily determined from the number of H→L and L→H transition events based on the same confined/mobile (L/H) classification using d_b_ = 60 nm. …”(p.30-p.32 of the revised manuscript; verbatim quote except “Figure 3 —figure supplement 3” in the revised manuscript is replaced by “Figure 3—figure supplement 3”).

8. (p. 8) The authors claim that the diffusion coefficient of NR2B in a mobile state in the condensed phase and that of NR2B in the dilute phase are "very close," but the reported values are ~0.47 μm2/s and ~0.61 μm2/s. The authors should clarify in what sense they are very close to each other.

Thanks for the comment. The essential message we intend to convey is that NR2B diffusion in the dilute phase is significantly closer to that of the mobile state in the condensed phase than that of the condensed phase as a whole (which is dominated by the confined state). In view of the complex (anomalous) nature of the diffusion processes in question and the additional model-independent analysis we describe above, we also recognize that values for condensed-phase mobile-state diffusion coefficient are model-dependent estimates. Nonetheless, the observation of significantly different concomitant time scales in the condensed phase is robust. Taking all these nuanced conceptual and quantitative considerations into account, we have now rephrased the pertinent statements as: “The fitted diffusion coefficient of presumed simple diffusion of NR2B in the dilute phase of the present PSD condensate is 0.47±0.12 μm^2^/s, which is close to that of NR2B alone tethered to SLB (0.61±0.04 μm^2^/s) (Figure 3E). The quantitative similarity of these two estimated diffusion coefficients is particularly striking in view of the much lower apparent overall diffusion coefficient for subdiffusion of NR2B in the PSD condensed phase (*D* = 0.014±0.001 μm^2^/s with α = 0.74±0.03; see Figure 3E)” (p.9 of the revised manuscript) and “the diffusion coefficient for the mobile fraction of molecules in the condensed phase may not be dramatically different from that in the dilute phase (*D*_m_ = 0.17 μm^2^/s vs *D*_d_ = 0.47 μm^2^/s), indicating that the landscape governing mobile-state diffusion in the condensed phase is only moderately rugged” (p.13 of the revised manuscript).

9. (p. 8) The authors mention that they did not observe any obvious hinderance against motions when molecules cross the phase boundaries (and hence assume that the flux is simply the diffusion coefficient multiplied by molecular density), but there are reports on the diffusion barrier at the boundaries (see 10.1038/nature22989 and 10.3390/ncrna5040050). In my opinion, the barrier makes sense, as the surface tension will break the symmetry between one side and the other side of the boundary. Can the authors comment on this?

Thanks for the comment. We recognize that boundary barrier widely exists in phase separation systems. We analysed potential barrier functions of droplet boundaries by monitoring whether motions of individual molecules are obviously speeded up or slowed down when crossing the phase boundaries. We reckon that any obvious speeded up or slowed down upon crossing would lead to accumulation or depletion of molecules at the phase boundaries. Accordingly, we have performed new measurements to address this question for our system. As indicated in Figure 3 —figure supplement 2, we did not detect obvious level of molecule enrichment or depletion near the boundary of each condensed patch (i.e., there is a sharp and smooth decrease in the boundary between the condensed phase and dilute phase; see the line plot analysis shown in Figure 3 —figure supplement 2B1-B3). Nonetheless, the reviewer’s comment is well taken. The boundary between the dilute and condensed phase in biological condensates is indeed an important research question, which we intend to look into in the future. We have now included the following statement now included in the revised text: “Because we did not observe any obvious hindrance against motions when molecules cross the phase boundaries (Figure 3—figure supplement 2), the energy barrier at the interface between the condensed and dilute phases is likely not large in this system (Brangwynne et al., 2011; Feric et al., 2016), though diffusion barriers at phase boundaries were reported in other systems (Peng and Weber, 2019; Strom et al., 2017)”(p.11 of the revised manuscript) [the latter two references are those mentioned by the reviewer under point #9].

10. (p. 9) The theoretical value for the enrichment fold is 62.8 and its experimental value is ~61, which may look astonishing, but the "theoretical" value is not calculated from scratch. It is based on the parameters that fit the data, so I think it is unsurprising that they obtain very similar values. I would be surprised if there was a big discrepancy.

Thanks for the comment. The experimental value was calculated with the localization density ratio of stochastically emitted fluorescent molecules in condensed phase region and dilute phase region. It can therefore be view as distributions of molecules over the time duration of the imaging process. In contrast, the theoretical value of the enrichment fold was calculated from the *kinetic* parameters extract from the single molecule tracking algorithm with a diffusion model with the approximate relation given in eq.1. Thus, these two values were derived independently. We have now clarified the situation by including the following sentence on p.11 of the revised manuscript: “It follows that the enrichment fold of NR2B in the condensed phase estimated via the approximate relation in eq. 1 by using these experimental kinetic parameters is *D*d/Pm*D*m = 62.8. Since this kinetics-estimated EF is very close to the equilibrium value of ~61 derived independently from experimentally observed localizations (Figure 1B), we regard eq. 1 as empirically validated for the present PSD system, although the relationship between kinetic and thermodynamic properties of diffusive systems (Berry et al., 2018) can be rather complex in general for rugged energy landscapes (Banerjee et al., 2014; Zwanzig, 1988).”

11. (p. 9) Again, they report the "remarkable" similarity between the diffusion coefficient for the mobile fraction in the condensed phase and that in the dilute phase, but I think the difference (0.17 vs 0.47 μm2/s) can be considered significant, depending on your perspective.

Thanks for the comments. We should indeed be more cautious in using exaggeratory adjectives like “remarkable”. We have removed it in the revised manuscript. In addition, we have added that we compared the diffusion coefficient for the mobile fraction in the condensed phase and that in the dilute phase (0.17 vs 0.47 μm^2^/s) with the background of the measured apparent diffusion coefficient ~0.014 μm^2^/s. As partially quoted above in our response to this reviewer’s point #8: “although concentrations of molecules in the condensed phase are much higher than those in the dilute phase, the diffusion coefficient for the mobile fraction of molecules in the condensed phase may not be dramatically different from that in the dilute phase (*D*_m_ = 0.17 μm^2^/s vs *D*_d_ = 0.47 μm^2^/s), indicating that the landscape governing mobile-state diffusion in the condensed phase is only moderately rugged … the apparent overall diffusion constant of NR2B in the condensed phase is small at *D* = 0.014 μm^2^/s instead of 0.17 μm^2^/s estimated for the mobile NR2B fractions in the condensed phase” (p.13 of the revised manuscript).

12. The authors conducted a Monte Carlo simulation to obtain the simulated enrichment fold. Is it just for validation of the analytical formula? (If this is the case, I don't think it is necessary to include the simulation results.) Can we obtain any other useful information from the simulation?

Thanks for the comment. The Monte Carlo simulation also show us the simulated system reaches a diffusion equilibrium state that matches the theoretical prediction. In this regard, it is a self-consistency check necessary for building a foundation for the parameter setting in FRAP simulation in the last section of the manuscript. Moreover, the simulation results also exhibit minor deviations that likely arise from the peculiar shapes of some of the phase boundaries, which we have now noted in the text as an interesting aspect for future investigations into possible model refinements in the interpretation of tracking data. Please see discussion on p.11-p.12 of the revised manuscript.

13. In Figure 4A and the corresponding text, the authors seem to claim that "weak" interactions will lead to "small" networks, but I think that this statement can be misleading. It may be true that molecules are involved in small networks at a certain time point, but the interactions are transient and dynamic (as the authors mention), so molecules change their partners rapidly and on average, the molecules are involved in a large (if not system-spanning) network. Hence, the "size" of the network should be discussed with care, and I recommend the authors revise the manuscript accordingly.

This is a great point. Molecular networks in the condensed phase formed by weak interactions are more transient and dynamic compared to networks formed by strong interactions. It is ambiguous to describe the network complexity and dynamics with size. We have revised the description of percolated molecular networks accordingly by eliminating references to size in Figure 4A and related text (p.15-p.16) in the revised manuscript.

14. (p. 11) The authors say that the fraction of dwell time is "directly proportional" to the binding affinity and avidity, which I don't think is a mathematically precise statement. Did they mean "directly dependent on the two factors"?

Thanks for drawing this to our attention. We have now revised the sentence to (p.16 of the revised manuscript): “The fraction of time that a molecule stays on the network is correlated with its binding affinity (i.e., the off-rate of the molecule from the network, a value directly related to the dissociation constant of the binding) and avidity (a combination of the available binding sites in the vicinity of the molecule, a parameter related to valency of the molecular interactions in the system and the binding affinity of each individual binding site) between the molecule and the network”.

15. If I understand correctly, the FUS experiment was conducted in 3D, unlike the NR2B experiment, where the molecules are attached to the membrane. As the system becomes three-dimensional, the "displacement" is now a 2D-projected value, and we need to devise a way to convert it to a 3D value. Do the authors consider this point? If so, please provide a description of their conversion method.

Thanks for your comment. The PrLD experiments were indeed performed in 3D condensates and molecules diffused in 3D space. For our experiments, we simply recorded and measured diffusion in 2D projections in each given imaging focal plane. One possible way to convert it back to 3D diffusion is to consider the diffusion is isotropic and independent in each direction in 3D. According to the principle of random walk, the mean square displacement is related to the diffusion coefficient:

〈*r*^2^〉 = 2*nDt*

where *n* is the number of spatial dimensions of the displacements. If diffusion is isotropic and independent of direction, the value of the diffusion coefficient in 3D deduced from this relation would be identical to that measured in 2D projection. The reason that we didn’t expand this 2D result into 3D is that we only recorded the lateral (x and y direction) positions of each single molecule. We could not measure the motion on the z direction as the molecules rapidly moved out of the focal plane of our imaging system. In other words, the efficiency of tracking single molecular motion in a 3D system is much lower than that on a 2D supported membrane system, as only those molecules with their motions stayed within the thin focal plane could be successfully tracked. Addition of a detailed discussion of these aspects of 2D vs 3D diffusion is beyond the scope of the present work. The reviewer’s point is well taken nonetheless. To clarify how we analysed data from our FUS experiment, we have now added the following sentence to the caption for Figure 4D: “Our analysis of diffusion data from the experiments depicted in this figure was based on 2D projections of 3D diffusion tracks”.

16. (p. 12) The authors say that phase separation of FUS PrLD took up to 12 hours to occur but do not show any data. Please include the data (microscope images or turbidity data).

We imaged the same PrLD sample in different time point after GB1 tag cleavage as shown in Figure 4—figure supplement 2. This supplementary figure is now referred to in the discussion on p.17 of the revised manuscript.

17. (p. 15) Can the authors comment on why they have greater error bars for simulated FRAP curves in Figure 5G? Is it because they have a smaller number of measurements (3 vs. 10)?

Thanks for the comment. The larger statistical uncertainties seen in Figure 5G of our originally submitted manuscript was caused by the smaller number of repeated simulations used for the results. Now, to be consistent with the analogous simulations in the same figure, we have replaced Figure 5G by a new result with 10 repeated simulations instead of 3.

Reviewer #2 (Recommendations for the authors):The central finding that the molecules tend to experience transiently confined states in the condensed phase is remarkable and important. This finding is reminiscent of transient "caging"/"trapping" dynamics observed in diverse other crowded and confined systems e.g., https://doi.org/10.1103%2FPhysRevLett.107.178103, https://doi.org/10.1038/s41467-019-10115-1, https://doi.org/10.1103%2FPhysRevLett.89.095704, https://doi.org/10.1103%2FPhysRevLett.92.178101, https://doi.org/10.1529%2Fbiophysj.106.092619, https://doi.org/10.1016%2Fj.bpj.2013.12.013. The authors may wish to comment on these conceptual connections to other systems that highlight the broader context of this fascinating finding; it might motivate others to bring theoretical and analytical approaches developed to understand these other systems to bear on condensates, which would be valuable to the field.Related to the previous point: it would be interesting to see not just the distribution of displacements, but also the distribution of times spent in the confined state and mobile state. Given the experimental results, the authors likely already have these data. The functional form of this distribution is known to reflect the physics underlying the trapping behavior and transitions between the two states (see e.g., https://doi.org/10.1016/0370-1573(90)90099-N).Also related to the previous point: it is very surprising to see the authors interpret the single-molecule motion as being 'normal' diffusion (within the context of a two-state diffusion model), instead of analyzing their data within the context of continuous time random walks or anomalous diffusion, which is generally known to arise from transient trapping in crowded/confined systems (again see e.g., https://doi.org/10.1103%2FPhysRevLett.107.178103, https://doi.org/10.1038/s41467-019-10115-1, https://doi.org/10.1103%2FPhysRevLett.89.095704, https://doi.org/10.1103%2FPhysRevLett.92.178101, https://doi.org/10.1529%2Fbiophysj.106.092619, https://doi.org/10.1016%2Fj.bpj.2013.12.013). It is not clear that interpreting the results within the context of simple diffusion is appropriate, given their general finding of the two confined and mobile states. Such a process of transient trapping/confinement is known to lead to transient subdiffusion at short times and then diffusive behavior at sufficiently long times. There is a hint of this in the inset to Figure 3, but these data need to be shown on log-log axes to be clearly interpreted. I encourage the authors to think more carefully and critically about the nature of the diffusive model to be used to interpret their results.

We thank the reviewer for the references and comments. As stated throughout this response letter, especially in our specific replies to this reviewer (reviewer 2)’s points 1, 2, 3 and reviewer 1’s point 7 above, we have conducted extensive new analyses during revision and have modified the manuscript to put the models that we applied in appropriate scientific context of prior efforts on anomalous diffusion. As stated above, we are of the view that the semi-quantitative analysis that we provided in the revised manuscript is adequate for our present purposes, and that insights offered by more refined modeling approaches will be useful for future extension of our investigations.

All the references mentioned above by this reviewer (Akimoto et al., 2011; Berezhkovskii et al., 2014; Bhattacharjee and Datta, 2019; Bouchaud and Georges, 1990; Saxton, 2007; Weeks and Weitz, 2002; Wong et al., 2004) are now cited in the revised manuscript and used in our extended discussion as they are indeed useful for providing a proper context of our work to the *eLife* readership. These additions are listed in our reply to comments from both reviewers. For instance, the following passage citing some of these references is now included in the revised manuscript:

“The above result indicated that NR2B in the condensed phase did not undergo homogeneous diffusion motions as one might expect for molecules in certain crowded condensed phases (Condamin et al., 2008; Höfling and Franosch, 2013; Saxton, 2007; Woringer and Darzacq, 2018). Instead, the motions of NR2B in the condensed phase resemble previously studied crowded and confined systems with transient caging/trapping dynamics (Akimoto et al., 2011; Bhattacharjee and Datta, 2019; Weeks and Weitz, 2002; Wong et al., 2004). Comparisons with data from a set of model experimental systems of FUS-Shank3 chimeric proteins (see Figure 4) suggest that the transient kinetic trapping of NR2B (transiently remaining very close to a stationary position, i.e., very slow diffusion) in the PSD condensed phase is likely due to its relatively strong binding to certain parts of a percolated PSD network with slow dynamics. Accordingly, NR2B molecules in the condensed phase are expected to undergo subdiffusive motions. Indeed, our single molecule tracking data of NR2B motion in the condensed PSD phase (see Figure 2E for an example) fits well with typical behaviors of subdiffusion (Bouchaud and Georges, 1990; Condamin et al., 2008; Netz and Dorfmüller, 1995; Saxton, 2007)” (p.7-p.8 of the revised manuscript).

References:

Abyzov, A., Blackledge, M., and Zweckstetter, M. (2022). Conformational Dynamics of Intrinsically Disordered Proteins Regulate Biomolecular Condensate Chemistry. Chemical Reviews *122*, 67196748.

Akimoto, T., Yamamoto, E., Yasuoka, K., Hirano, Y., and Yasui, M. (2011). Non-Gaussian Fluctuations Resulting from Power-Law Trapping in a Lipid Bilayer. Physical Review Letters *107*, 178103. Banerjee, S., Biswas, R., Seki, K., and Bagchi, B. (2014). Diffusion on a rugged energy landscape with spatial correlations. The Journal of chemical physics *141*, 124105.

Berezhkovskii, A.M., Dagdug, L., and Bezrukov, S.M. (2014). Discriminating between anomalous diffusion and transient behavior in microheterogeneous environments. Biophysical journal *106*, L09L11.

Berry, J., Brangwynne, C., and Haataja, M. (2018). Physical Principles of Intracellular Organization via Active and Passive Phase Transitions. Reports on Progress in Physics *81*, 046601.

Bhattacharjee, T., and Datta, S.S. (2019). Bacterial hopping and trapping in porous media. Nature Communications *10*, 2075.

Bouchaud, J.P., and Georges, A. (1990). Anomalous diffusion in disordered media: Statistical mechanisms, models and physical applications. Phys. Rep. *195*, 127-293.

Brangwynne, C.P., Mitchison, T.J., and Hyman, A.A. (2011). Active liquid-like behavior of nucleoli determines their size and shape in *Xenopus laevis* oocytes. Proc Natl Acad Sci U S A *108*, 4334-4339. Cates, M.E., and Tailleur, J. (2015). Motility-Induced Phase Separation. Annual Review of Condensed Matter Physics *6*, 219-244.

Cho, W.-K., Spille, J.-H., Hecht, M., Lee, C., Li, C., Grube, V., and Cisse, I.I. (2018). Mediator and RNA polymerase II clusters associate in transcription-dependent condensates. Science *361*, 412-415. Condamin, S., Tejedor, V., Voituriez, R., Bénichou, O., and Klafter, J. (2008). Probing microscopic origins of confined subdiffusion by first-passage observables. Proceedings of the National Academy of Sciences *105*, 5675-5680.

Feric, M., Vaidya, N., Harmon, T.S., Mitrea, D.M., Zhu, L., Richardson, T.M., Kriwacki, R.W., Pappu, R.V., and Brangwynne, C.P. (2016). Coexisting Liquid Phases Underlie Nucleolar Subcompartments. Cell *165*, 1686-1697.

Garcia, D.A., Johnson, T.A., Presman, D.M., Fettweis, G., Wagh, K., Rinaldi, L., Stavreva, D.A., Paakinaho, V., Jensen, R.A., and Mandrup, S. (2021). An intrinsically disordered region-mediated confinement state contributes to the dynamics and function of transcription factors. Molecular cell *81*, 1484-1498. e1486.

Guilhas, B., Walter, J.-C., Rech, J., David, G., Walliser, N.O., Palmeri, J., Mathieu-Demaziere, C., Parmeggiani, A., Bouet, J.-Y., Le Gall, A.*, et al.* (2020). ATP-Driven Separation of Liquid Phase Condensates in Bacteria. Molecular Cell *79*, 293-303.e294.

Höfling, F., and Franosch, T. (2013). Anomalous transport in the crowded world of biological cells.

Reports on Progress in Physics *76*, 046602.

Joo, S., Durang, X., Lee, O.-c., and Jeon, J.-H. (2020). Anomalous diffusion of active Brownian particles cross-linked to a networked polymer: Langevin dynamics simulation and theory. Soft Matter *16*, 9188-9201.

Kent, S., Brown, K., Yang, C.-h., Alsaihati, N., Tian, C., Wang, H., and Ren, X. (2020). Phase-separated transcriptional condensates accelerate target-search process revealed by live-cell single-molecule imaging. Cell reports *33*, 108248.

Kuhn, T., Hettich, J., Davtyan, R., and Gebhardt, J.C.M. (2021). Single molecule tracking and analysis framework including theory-predicted parameter settings. Scientific Reports *11*, 9465.

Moon, S.L., Morisaki, T., Khong, A., Lyon, K., Parker, R., and Stasevich, T.J. (2019). Multicolour singlemolecule tracking of mRNA interactions with RNP granules. Nature Cell Biology *21*, 162-168. Muñoz-Gil, G., Romero-Aristizabal, C., Mateos, N., Campelo, F., de Llobet Cucalon, L.I., Beato, M.,

Lewenstein, M., Garcia-Parajo, M.F., and Torreno-Pina, J.A. (2022). Stochastic particle unbinding modulates growth dynamics and size of transcription factor condensates in living cells. Proceedings of the National Academy of Sciences *119*, e2200667119.

Muñoz-Gil, G., Volpe, G., Garcia-March, M.A., Aghion, E., Argun, A., Hong, C.B., Bland, T., Bo, S., Conejero, J.A., and Firbas, N. (2021). Objective comparison of methods to decode anomalous diffusion. Nature communications *12*, 6253.

Netz, P.A., and Dorfmüller, T. (1995). Computer simulation studies of anomalous diffusion in gels:

Structural properties and probe‐size dependence. The Journal of chemical physics *103*, 9074-9082.

Peng, A., and Weber, S.C. (2019). Evidence for and against Liquid-Liquid Phase Separation in the Nucleus. Non-coding RNA *5*, 50.

Saxton, M.J. (2007). A biological interpretation of transient anomalous subdiffusion. I. Qualitative model. Biophysical journal *92*, 1178-1191.

Song, D., Jo, Y., Choi, J.-M., and Jung, Y. (2020). Client proximity enhancement inside cellular membrane-less compartments governed by client-compartment interactions. Nature Communications *11*, 5642.

Song, J., Gomes, G.-N., Shi, T., Gradinaru, C.C., and Chan, H.S. (2017). Conformational heterogeneity and FRET data interpretation for dimensions of unfolded proteins. Biophysical journal *113*, 10121024.

Strom, A.R., Emelyanov, A.V., Mir, M., Fyodorov, D.V., Darzacq, X., and Karpen, G.H. (2017). Phase separation drives heterochromatin domain formation. Nature *547*, 241-245.

Tang, A.-H., Chen, H., Li, T.P., Metzbower, S.R., MacGillavry, H.D., and Blanpied, T.A. (2016). A transsynaptic nanocolumn aligns neurotransmitter release to receptors. Nature *536*, 210-214. Taylor, N.O., Wei, M.-T., Stone, H.A., and Brangwynne, C.P. (2019). Quantifying Dynamics in PhaseSeparated Condensates Using Fluorescence Recovery after Photobleaching. Biophysical Journal *117*, 1285-1300.

Viswanathan, G.M., Raposo, E., and Da Luz, M. (2008). Lévy flights and superdiffusion in the context of biological encounters and random searches. Physics of Life Reviews *5*, 133-150.

Wang, X., Chen, Y., and Deng, W. (2020). Strong anomalous diffusion in two-state process with Lévy walk and Brownian motion. Physical Review Research *2*, 013102.

Weeks, E.R., and Weitz, D. (2002). Properties of cage rearrangements observed near the colloidal glass transition. Physical review letters *89*, 095704.

Wong, I.Y., Gardel, M.L., Reichman, D.R., Weeks, E.R., Valentine, M.T., Bausch, A.R., and Weitz, D.A. (2004). Anomalous Diffusion Probes Microstructure Dynamics of Entangled F-Actin Networks.

Physical Review Letters *92*, 178101.

Woringer, M., and Darzacq, X. (2018). Protein motion in the nucleus: from anomalous diffusion to weak interactions. Biochemical Society Transactions *46*, 945-956.

Zhang, Z., and Chan, H. (2012). Transition paths, diffusive processes, and preequilibria of protein folding. Proceedings of the National Academy of Sciences of the United States of America *109*, 20919-20924.

Zwanzig, R. (1988). Diffusion in a rough potential. Proceedings of the National Academy of Sciences of the United States of America *85*, 2029-2030.